# Constraint-Conditioned Actor-Critic for Offline Safe Reinforcement Learning

**Zijian Guo**[1]**, Weichao Zhou**[2]**, Shengao Wang**[1]**, Wenchao Li**[1,2]
[1]Division of Systems Engineering, Boston University
[2]Department of Electrical and Computer Engineering, Boston University
`{zjguo, zwc662, wsashawn, wenchao}@bu.edu`

## Abstract

Offline safe reinforcement learning (OSRL) aims to learn policies with high rewards while satisfying safety constraints solely from data collected offline. However, the learned policies often struggle to handle states and actions that are not present or out-of-distribution (OOD) from the offline dataset, which can result in violation of the safety constraints or overly conservative behaviors during their online deployment. Moreover, many existing methods are unable to learn policies that can adapt to varying constraint thresholds. To address these challenges, we propose constraint-conditioned actor-critic (CCAC), a novel OSRL method that models the relationship between state-action distributions and safety constraints, and leverages this relationship to regularize critics and policy learning. CCAC learns policies that can effectively handle OOD data and adapt to varying constraint thresholds. Empirical evaluations on the `DSRL` benchmarks show that CCAC significantly outperforms existing methods for learning adaptive, safe, and high-reward policies. The code is available at https://github.com/BU-DEPEND-Lab/CCAC.

## 1 Introduction

Offline safe reinforcement learning (OSRL) seeks to learn policies that maximize rewards while adhering to safety constraints solely based on offline datasets. OSRL brings together key aspects of safe RL (Gu et al., 2024) and offline RL (Levine et al., 2020; Prudencio et al., 2023). It adds constraints to ensure that the learned policy meets safety requirements (Lin et al., 2024; Cao et al., 2024; Guo et al., 2024), while also learning from offline datasets to avoid expensive and potentially hazardous online environment interactions (Bharadhwaj et al., 2020; Shi et al., 2021; Kondrup et al., 2023; Zhou et al., 2023). However, it also inherits the challenges from these two areas.

A major challenge in offline RL is distribution shifts (Levine et al., 2020): the behavior of a policy trained offline can become unpredicatble when encountering states and actions unseen or out-of-distribution (OOD) from the offline datasets during deployment. To mitigate this issue, many existing offline reinforcement learning methods limit the policy's deviation from the (unknown) behavior policy that generated the offline dataset or underestimate the critic's value. (Lyu et al., 2022; Kostrikov et al., 2021; Garg et al., 2023; Kumar et al., 2020). However, such approaches often result in overly conservative policies. Additionally, the theoretical foundation of many offline RL methods is based on the assumption that the offline dataset is collected with a single behavior policy. However, in real-world tasks, a dataset is more likely to be collected from multiple and diverse behavior policies. For example, the training data for self-driving vehicles are often collected by multiple human drivers whose policies are different from one another.

As for safe RL, a key challenge is the lack of adaptability to varying constraint thresholds after training (Liu et al., 2022). Here, "varying" means that during deployment, trained agents are assigned a constraint threshold that may differ for each rollout or can change dynamically over time. This limitation stems from the common adoption of a constrained optimization formulation (Xu et al., 2022; Hong et al., 2024; Guan et al., 2024; Zheng et al., 2023), which typically involves a single, fixed constraint threshold during training. However, in many real-world safety-critical applications, the ability to adapt to different or even unseen constraint thresholds is essential. For example, a self-driving vehicle must adjust its behavior to meet varying safety requirements, such as altering

its speed in response to different speed limits on the road. While some existing methods tackle the problem of varying constraint thresholds using supervised learning (Liu et al., 2023b; Zhang et al., 2023b) or trajectory optimization (Lin et al., 2023), they are limited in their abilities to generalize to potential unseen constraint thresholds between training and deployment.

In this work, we tackle these challenges simultaneously by learning a safe, high-reward policy from offline datasets that can adapt to varying constraint thresholds while effectively managing distribution shifts. Our key idea is to model the distribution of states and actions and uncover the relationships between behaviors and cost constraints from the offline dataset. We introduce a novel method called constraint-conditioned actor-critic (CCAC), which combines a constraint-conditioned variational autoencoder (CVAE) and a constraint-conditioned classifier to learn this relationship. The CVAE facilitates data augmentation by generating new training samples that align with the learned behavior-constraint relationship across different constraint thresholds. Meanwhile, the constraint-conditioned classifier identifies OOD data within the generated samples, allowing for a selectively conservative update of the critics and actor. This structure enables CCAC to train a policy that can effectively handle OOD data and adapt to varying constraint thresholds during online deployment. Our main contributions are summarized as follows.

- We propose a novel approach for learning adaptive, safe, and high-reward policies in OSRL. To our knowledge, this is the first value-based method that can achieve zero-shot adaptation to varying constraints in the OSRL setting.

- We introduce a novel constraint-conditioned generative model and binary classifier for generating state-action pairs and detecting OOD data, respectively. We demonstrate that these components can be used to effectively regularize the learning of both the critics and the actor.

- We conduct comprehensive experiments to show that (i) CCAC outperforms state-of-the-art baselines both in safety and task performance by a large margin, and (ii) CCAC can achieve high rewards while generalizing to varying constraint thresholds without re-training the policy.

## 2 RELATED WORK

**Constraint satisfaction in RL.** Many methods across different paradigms have been proposed to achieve constraint satisfaction. In offline safe RL, most of the works formulate the problem as a constrained optimization problem, including primal-dual methods that optimize iteratively the primal and dual problem (Xu et al., 2022; Hong et al., 2024) and stationary-distribution-correction-style methods that train policies via importance sampling (Polosky et al., 2022; Lee et al., 2022). (Zheng et al., 2023) breaks down the offline safe RL problem into separate components and solves them individually. However, all of these methods require a fixed constraint threshold. Several works convert the offline safe RL problem into sequential modeling and use transformers to learn conditioned policies (Liu et al., 2023b; Zhang et al., 2023b). Although they can generalize to different cost thresholds, they are sensitive to the rewards and costs they are conditioned on, e.g., combinations of high rewards and low costs that do not exist in the dataset can degrade the performance. (Lin et al., 2023) solves the offline safe RL problem from a trajectory optimization perspective, but it assumes a single behavior policy and constrains its policy to stay close to the behavior policy. For online safe RL, (Zhang et al., 2021; Sootla et al., 2022) learns policies by maximizing the reward with constraint-related information included in the states. (Yao et al., 2024b) also learns constraint-conditioned policies, but they consider constant thresholds. Moreover, it is nontrivial to tackle the issue of distribution shifts when switching from an online to an offline setting.

**Distribution shifts in offline (safe) RL.** In the offline setting, a major challenge is to address the distribution shifts caused by OOD states and actions. Various methods have been proposed to penalize the divergence between the learned policy and the behavior policy, using metrics such as Maximum Mean Discrepancy (MMD) (Kumar et al., 2019), Fisher divergence (Kostrikov et al., 2021) and KL divergence (Nair et al., 2020; Wu et al., 2019). Generative models, e.g., VAEs and GANs, are frequently used to generate actions based on states (Kumar et al., 2019; Chen et al., 2022; Wu et al., 2022; Zhang et al., 2023a; Liu et al., 2024) in order to regularize critics and policy learning and similar techniques are adopted when considering safety (Xu et al., 2022). However, the dependence of generating actions on the constraints is rarely explored. Although (Yao et al., 2024a) consider distribution shift based on constraints, they still focus on a single constraint threshold. While model-based RL methods can leverage learned dynamics models for prediction and planning

to address state distribution shifts (Kidambi et al., 2021; Yu et al., 2020; Diehl et al., 2022; Cho et al., 2024), the mitigation of OOD states effects is rarely explored in model-free methods. By conditioning on the constraint, we are able to address the distribution shifts of both OOD states and actions in a model-free manner.

## 3 PRELIMINARIES

**CMDP and offline safe RL.** We consider the Constrained Markov Decision Process (CMDP) model (Altman, 1998) defined by the tuple $(\mathcal{S}, \mathcal{A}, P, r, c, \gamma, \mu)$, where $\mathcal{S}$ is the state space, $\mathcal{A}$ is the action space, $P : \mathcal{S} \times \mathcal{A} \times \mathcal{S} \rightarrow [0, 1]$ is the (unknown) transition function, $r : \mathcal{S} \times \mathcal{A} \rightarrow [-R_{max}, R_{max}]$ is the reward function with $R_{max}$ the maximum reward, $c : \mathcal{S} \times \mathcal{A} \rightarrow [0, C_{max}]$ is the cost function with $C_{max}$ the maximum cost, $\gamma$ is the discounted factor, and $\mu : \mathcal{S} \rightarrow [0, 1]$ is the initial state distribution. Let $\pi : \mathcal{S} \times \mathcal{A} \rightarrow [0, 1]$ denote the policy; $\tau = \{s_t, a_t, c_t, r_t\}_{t=1}^T$ denote the trajectory with maximum episode length $T$; $R(\tau) = \sum_{t=1}^T \gamma^t r_t$ denote the reward return of the trajectory $\tau$; and $C(\tau) = \sum_{t=1}^T \gamma^t c_t$ denote the cost return. The goal is to find a policy that maximizes the reward return while limiting the cost return to a threshold $\epsilon$ from a fixed dataset $\mathcal{D} = \{\tau_i\}_{i=1}^N$:

$$\max_\pi \mathbb{E}_{\tau \sim \pi}\big[R(\tau)\big], \quad s.t. \quad \mathbb{E}_{\tau \sim \pi}\big[C(\tau)\big] \leq \epsilon. \tag{1}$$

**Actor-critic methods.** The actor-critic methods (Haarnoja et al., 2018; Stooke et al., 2020) solve Eq.(1) by maintaining two critics $Q_r, Q_c$ to respectively approximate the the Q-value for the reward $Q_r^\pi(s, a) \equiv \mathbb{E}_{\tau \sim \pi}[R(\tau)]$ and Q-value for the cost $Q_c^\pi(s, a) \equiv \mathbb{E}_{\tau \sim \pi}[C(\tau)]$ with $\tau = \{s, a, \dots\}$. The critics $Q_r, Q_c$ and actor $\pi$ are updated in an alternating manner.

$$\min_{Q_f} \mathbb{E}_{s,a \sim \mathcal{D}}\big[(Q_f(s, a) - \mathcal{T}^\pi Q_f(s, a))^2\big], f \in \{r, c\} \quad \text{(policy evaluation)}$$

$$\max_\pi \mathbb{E}_{s \sim \mathcal{D}, a \sim \pi}\big[Q_r(s, a)\big] \text{ s.t. } \mathbb{E}_{s \sim \mathcal{D}, a \sim \pi}\big[Q_c(s, a)\big] \leq \epsilon \quad \text{(policy improvement)} \tag{2}$$

The critics are updated via policy evaluation, by iterating the Bellman operator: $\mathcal{T}^\pi Q_f(s, a) = f(s, a) + \gamma P^\pi Q_f(s, a)$, where $f \in \{r, c\}$ and $P^\pi$ is the transition matrix associated with the policy: $P^\pi Q_f(s, a) = \mathbb{E}_{s' \sim P, a' \sim \pi}[Q_f(s', a')]$. The policy is improved by updating it towards actions that maximize the expected $Q_r$-values and whose $Q_c$-values satisfy the constraint threshold.

## 4 CONSTRAINT-CONDITIONED ACTOR-CRITIC

In this section, we introduce our proposed method as illustrated in Figure 1. In Section 4.1, we argue that the offline dataset can be viewed as collected by a behavior policy that chooses different actions depending on the constraint threshold. Building on this perspective, we focus on learning the dependence of the state-action pairs on the constraint thresholds within the offline dataset. In particular, we treat the state-action pairs that violate cost constraints as OOD. In Section 4.2, we introduce the constraint-conditioned VAE and constraint-conditioned classifier to model the OOD state-action distribution. In Sections 4.3 and 4.4, we provide details on how we apply the learned OOD distribution to regularize the learning of constraint-conditioned critics and actor to produce safe and high-reward policies.

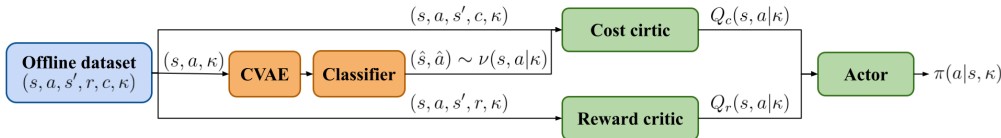

Figure 1: CCAC overview. During training, we sample $(s, a, s', r, c, \kappa)$ from the dataset and use $(s, a, \kappa)$ to train the CVAE and Classifier, which aim to learn a constraint-conditioned distribution of OOD state-action pairs, $\nu(s, a|\kappa)$. Specifically, we generate pairs $(\hat{s}, \hat{a})$ by sampling from the CVAE, and the pairs with a classifier score above 0.5 are considered OOD. Then these OOD pairs, along with the sampled $(s, a, s', c, \kappa)$, are used to update the cost critic, $Q_c(s, a|\kappa)$, while the sampled $(s, a, s', r, \kappa)$ are used to update the reward critic, $Q_r(s, a|\kappa)$. Finally, the critics are employed to update the policy $\pi(a|s, \kappa)$.

### 4.1 RETHINKING THE OFFLINE DATASET: A CONSTRAINT-CONDITIONED PERSPECTIVE

As mentioned in Section 2, many metrics have been proposed to measure the distribution shifts of states and actions in offline RL. However, those metrics often overlook the correlation between the constraint thresholds and the state-action distribution. We argue that this correlation is a significant aspect of offline datasets that can be exploited not only for characterizing the distribution shift but also for learning adaptive policies. In our method, we define the varying constraint threshold at each time-step $t$ as the cost budget, denoted as $\kappa_t$. Specifically, for each trajectory in the offline dataset, we set the cumulative cost of that trajectory as its initial cost budget, i.e., $\kappa_1 = \sum_{t=1}^{T} c_t$. The cost budget is then updated at each time-step based on the incurred cost, i.e., $\kappa_{t+1} = \kappa_t - c_t$, for $t = 1, 2, \ldots, T-1$. This formulation is similar to the notions of return-to-go (Chen et al., 2021) and cost-to-go (Liu et al., 2023b). Compared with existing safe RL methods that use a constant threshold, varying constraint thresholds provides better flexibility for the agent to adjust its behaviors, e.g., opting for more conservative actions as the cost budget decreases.

Considering the cost budget as the underlying cause of the behaviors in the dataset, the state-action distribution in the offline dataset $\mathcal{D} = \{\tau_i\}_{i=1}^{N}$ with $\tau = \{s_t, a_t, c_t, r_t, \kappa_t\}_{t=1}^{T}$ can be factorized as $d^{\pi_{\mathcal{D}}}(\kappa)d^{\pi_{\mathcal{D}}}(s|\kappa)\pi_{\mathcal{D}}(a|s, \kappa)$, where $\pi_{\mathcal{D}}(a|s, \kappa)$ is some constraint-conditioned behavior policy, $d^{\pi_{\mathcal{D}}}(\kappa)$ and $d^{\pi_{\mathcal{D}}}(s|\kappa)$ represent the distributions of encountered cost budgets and visited states by following $\pi_{\mathcal{D}}$, respectively. Given a cost budget $\kappa$, we consider a state-action pair $(s, a)$ to be OOD if taking the action $a$ at the state $s$ will lead to exceeding the cost budget $\kappa$, and otherwise $(s, a)$ is considered to be in-distribution (IND). Note that the OOD issue – when encountering states and actions unseen in the dataset – is more nuanced in the offline safe RL setting due to the need to ensure safety in addition to policy performance. Our treatment of regarding state-action pairs that exceed the cost budget as OOD provides a means of incorporating safety constraints into the policy optimization process. This enables the learned policy to avoid unsafe regions of the state-action space even if they are within the dataset distribution, while also adapting to unseen and varying cost budgets. As we will show in Section 4.2 and Section 4.3, this is done through a novel data augmentation and filtering process by a constraint-conditioned VAE and classifer, and an overestimation of the cost critic respectively. In Section 4.4, we offer further insights on how this overestimation can achieve a similar policy regularization effect as an underestimation of reward critic (Kumar et al., 2020; Lyu et al., 2022) to mitigate the OOD issue in offline RL.

### 4.2 DATA GENERATION AND CLASSIFICATION

We aim to learn a constraint-conditioned policy $\pi(a|s, \kappa)$ that can satisfy the varying cost budget $\kappa$ by favoring the IND state-action pairs while avoiding the OOD ones. Given a dataset, we can readily classify the data as OOD or not based on a cost budget $\kappa$. However, empirical results in Section 5.3 indicate that directly learning such a policy from the offline dataset is often limited by the size of the dataset. To overcome this, we propose to learn the OOD state-action distribution, which we denote as $\nu(s, a|\kappa)$, and use it to augment the dataset. Specifically, we train a generative model to generate diverse state-action pairs and train a binary classifier to identify OOD samples within the generated data. For the generative model, we propose a constraint-conditional variational autoencoder (CVAE) adapted from VAE (Kingma & Welling, 2013; Sohn et al., 2015), which models the distribution by transforming an underlying latent manifold. The objective is to optimize the following evidence lower bound (ELBO):

$$\max_{p,q} \mathbb{E}_{s,a,\kappa\sim\mathcal{D}}\Big[\mathbb{E}_{z\sim q(z|s,a,\kappa)}\big[p(s,a|z,\kappa)\big] - \beta D_{KL}\big(q(z|s,a,\kappa)||p(z|\kappa)\big)\Big] \quad (3)$$

where $q(z|s, a, \kappa)$ is the encoder that maps the tuple of state, action, and cost budget into the latent space, $p(s, a|z, \kappa)$ is the decoder that reconstructs the vector in latent space back into state-action pair, $p(z|\kappa)$ is the prior distribution, typically set to $\mathcal{N}(0, I)$, and $\beta$ is a hyperparameter. The first term represents the reconstruction loss, and the second term is the KL-divergence between the encoder output and the conditional prior of $z$. Previous works (Fujimoto et al., 2018; Kumar et al., 2019; Xu et al., 2022) also use conditional VAEs, i.e., VAEs conditioned on states, but they use them for sampling actions solely and do not consider constraints.

To generate state-action pairs, we can randomly sample $z$ from the prior distribution $p(z|\kappa)$ and pass them through the decoder. However, to determine which generated state-action pairs are OOD, additional steps are needed. We propose to train a constraint-conditioned binary classifier $h(s, a|\kappa)$

to distinguish IND and OOD state-action pairs using the binary-cross-entropy loss:

$$\min_h \mathbb{E}_{s,a,\kappa \sim \mathcal{D}} \Big[ -y \log\big(h(s,a|\kappa)\big) - (1-y)\log\big(1 - h(s,a|\kappa)\big) \Big] \tag{4}$$

where $y$ is the label of the tuple $(s,a,\kappa)$. We label a state-action pair sampled from datasets by checking its cost budget. For example, during each training iteration, we randomly sample a batch of $(s,a,\kappa)$ and a cost budget $\bar{\kappa}$ from $\mathcal{D}$. We then assign labels: $y = 0$ (IND) if $\kappa \leq \bar{\kappa}$ and $y = 1$ (OOD) otherwise. Additionally, we include reconstructed state-action pairs $(\hat{s}, \hat{a})$ obtained from the CVAE in the training of the classifier, assigning them the same labels as their original counterparts $(s,a,\kappa)$. This is based on two mild assumptions: (i) the reconstructed data should closely resemble the real data, which is the goal of CVAE, and (ii) taking similar actions in similar states leads to a similar number of constraint violations. The reconstructed data can be seen as small perturbations of the original data. Consequently, the batch data for classifier training is augmented to include both $(s,a,\bar{\kappa},y)$ and $(\hat{s},\hat{a},\bar{\kappa},y)$, enhancing the effectiveness of the training process. Considering $h(s,a|k)$ as the probability of $(s,a,k)$ being OOD, the OOD state-action pair distribution $\nu(s,a|\kappa)$ can be proportional to $\mathbb{E}_{z \sim p(z|\kappa)}[p(s,a|z,\kappa)h(s,a|\kappa)]$.

### 4.3 Constraint-Conditioned Critics

**Constraint-conditioned cost critic.** To achieve safety, we aim to overestimate the cost critic of OOD state-action pairs within bounds. Our choice is to learn a constraint-conditioned cost critic that encourages the expected $Q_c$-values under a particular distribution of state-action pairs, i.e., the distribution of the OOD state-action pairs $\nu(s,a|\kappa)$, to be no less than a specified threshold $\epsilon$ while minimizing the Bellman error, which yields the following objective:

$$\min_{Q_c} \mathbb{E}_{s,a,\kappa \sim \mathcal{D}} \big[ (Q_c(s,a|\kappa) - \mathcal{T}^\pi Q_c(s,a|\kappa))^2 \big] , \text{ s.t. } \mathbb{E}_{s,a \sim \nu, \kappa \sim \mathcal{D}}[Q_c(s,a|\kappa)] \geq \epsilon \tag{5}$$

Intuitively, $Q_c$-values of the state-action pairs that are well-supported by the data distribution $\pi_\mathcal{D}(s,a|\kappa)$, will be pushed down to comply with the Bellman backup by the Bellman error term. Conversely, those OOD and unsafe state-action pairs' $Q_c$-values will be pushed up. The overestimation can extend to IND state-action pairs that are near the boundary of IND and OOD, causing their $Q_c$-values to be higher than their true values. However, empirically, this does not show a noticeable impact on performance as it tends to lead to a mildly conservative policy that avoids risky actions.

In order to solve Eq.(5), we use the primal-dual method (Chow et al., 2018; Stooke et al., 2020) by introducing a dual variable, i.e., the Lagrangian multiplier $\lambda_c$, and transforming it into an unconstrained optimization problem:

$$\max_{\lambda_c \geq 0} \min_{Q_c} \mathbb{E}_{s,a,\kappa \sim \mathcal{D}} \big[ (Q_c(s,a|\kappa) - \mathcal{T}^\pi Q_c(s,a|\kappa))^2 \big] - \lambda_c(\mathbb{E}_{s,a \sim \nu, \kappa \sim \mathcal{D}}[Q_c(s,a|\kappa)] - \epsilon) \tag{6}$$

Now, we show that we can make $Q_c$-values of the OOD state-action pairs greater than $\epsilon$ with appropriate $\lambda_c$ when updating the cost critic by Eq.(5):

**Theorem 4.1.** *For any $\nu(s,a|\kappa)$ with supp $\nu \subset$ supp $\pi_\mathcal{D}$, $\forall \kappa \in \mathcal{D}$, $(s,a) \in \mathcal{D}_{ood}$, by iterating Eq.(5), the cost critic obtained $\hat{Q}_c^\pi(s,a|\kappa)$ satisfies:*

$$\hat{Q}_c^\pi(s,a|\kappa) = Q_c^\pi(s,a|\kappa) + \frac{\lambda_c}{2} \cdot \Big[ (I - \gamma P^\pi)^{-1} \frac{\nu(s,a|\kappa)}{d^{\pi_\mathcal{D}}(s,a|\kappa)} \Big]$$

*where $\lambda_c$ is the weight, $\mathcal{D}_{ood}$ is the set that contains all the OOD data generated by $\nu(s,a|\kappa)$, $d^{\pi_\mathcal{D}}(s,a|\kappa) = d^{\pi_\mathcal{D}}(s|\kappa)\pi_\mathcal{D}(a|s,\kappa)$ is the marginal state-action distribution, and if we choose $\lambda_c \geq \max\{2\max_{s,a,\kappa}\big[\frac{d^{\pi_\mathcal{D}}(s,a|\kappa)}{\nu(s,a|\kappa)}\big](\epsilon - Q_c(s,a|\kappa)(I - \gamma P^\pi)), 0\}$ then we can get $\hat{Q}_c^\pi(s,a|\kappa) \geq \epsilon$.*

The proof can be found in Appendix A. Note that $I - \gamma P^\pi$ is the inverse of the state occupancy matrix with non-negative entries and $\pi_\mathcal{D}(s,a|\kappa) > 0$ since the behavior policies can be sub-optimal, e.g., possible to violate constraints. One can set the constraint threshold $\epsilon$ to be greater than the maximum value of $\kappa$ in the offline datasets to ensure that the $Q_c$-values of OOD state-action pairs are sufficiently overestimated.

**Constraint-conditioned reward critic.** We learn a constraint-conditioned reward critic by minimizing the Bellman error without any additional terms:

$$\min_{Q_r} \mathbb{E}_{s,a,\kappa \sim \mathcal{D}} \big[ (Q_r(s,a|\kappa) - \mathcal{T}^\pi Q_r(s,a|\kappa))^2 \big] \tag{7}$$

Although directly applying the Bellman operator is known to suffer from OOD actions, overestimating the $Q_c$-values can avoid this issue when learning the policy.

## 4.4 CONSTRAINT-CONDITIONED ACTOR

To maintain consistency with the reward and cost critics, we train a constraint-conditioned actor by maximizing the reward while satisfying the constraints as follows.

$$\max_\pi \mathbb{E}_{s,\kappa\sim\mathcal{D},a\sim\pi(a|s,\kappa)}\left[Q_r(s,a|\kappa)\right], \text{ s.t. } \mathbb{E}_{s\sim\mathcal{D},a\sim\pi(a|s,\kappa)}\left[Q_c(s,a|\kappa)\right] \leq \kappa, \forall\kappa\in\mathcal{D} \quad (8)$$

Without conditioning $s$ on $\kappa$, Eq.(8) simplifies to the standard policy improvement in Eq.(2) and the constraint regarding the minimum $\kappa$ in $\mathcal{D}$ will dominate the other constraints. Although our definition of OOD differs from the typical definition of OOD (unseen state-actions in the dataset), our approach can still effectively mitigate this issue via the overestimation of $Q_c(s,a|\kappa)$ to achieve policy regularization. By overestimating the $Q_c(s,a|\kappa)$, we can naturally avoid high-cost actions as they will violate the constraint in Eq.(8). Additionally, this overestimation induces policy regularization, which is a common approach in offline RL to address the OOD issue by penalizing deviations of the learned policy from the behavior policy (Kumar et al., 2020; Lyu et al., 2022; Kostrikov et al., 2021). For example, CQL (Kumar et al., 2020) applies a conservative (underestimation of) $Q_r(s,a)$ to regularize the learned policy to stay close to the behavior policy. It is proved in Dual RL (Sikchi et al., 2023) that adding such a constraint on value functions is equivalent to adding a $f$-divergence between the learned policy and behavior policy. Leveraging the dual relationship between $Q_r(s,a|\kappa)$ and $Q_c(s,a|\kappa)$ (Paternain et al., 2019), our method achieves a similar policy regularization effect by overestimating $Q_c(s,a|\kappa)$, thereby keeping the learned policy close to the behavior policy. Thus, when learning the $Q_r(s,a|\kappa)$, our method does not need an additional explicit regularization. Empirically, we validate this in Appendix D.4 by replacing the reward function learning part in CCAC with an existing explicit regularization method.

Similar to solving Eq.(5), we also use the primal-dual method to solve Eq.(8):

$$\min_{\lambda_a\geq 0}\max_\pi \mathbb{E}_{s,\kappa\sim\mathcal{D},a\sim\pi}\left[Q_r(s,a|\kappa)\right] - \sum_{\kappa\in D}\lambda_a(\kappa)(\mathbb{E}_{s\sim\mathcal{D},a\sim\pi}\left[Q_c(s,a|\kappa)\right] - \kappa) \quad (9)$$

where $\lambda_a(\kappa)$ are the Lagrangian multipliers, which are parameterized by networks and tuned automatically. The overall method is summarized in Algorithm 1 in Appendix C.1. The training of OOD generation and detection, and that of the RL components can be performed separately. We start by training the CVAE and classifier till convergence. Then, we fix them and proceed to train the critics and actor. Training all components simultaneously can cause instability, as an inadequately trained OOD generation and detection module may produce incorrect OOD state-action pairs, ultimately leading to the failure of the critics and actor training. During evaluation, the initial cost budget is set arbitrarily for each rollout and updated at every time-step based on the costs incurred on the fly. A well-trained policy is expected to adjust its behavior accordingly.

## 5 EXPERIMENTS

In this section, we evaluate our method in multiple tasks to answer the following questions:

**Q1** Can CCAC learn safe and high-reward policies from offline datasets?

**Q2** Can CCAC achieve zero-shot adaption to different cost budgets?

**Q3** What is the importance of the OOD detection component in CCAC?

**Tasks.** The `Bullet-Safety-Gym` (Gronauer, 2022) and `Safety-Gymnasium` (Ji et al., 2023) are public benchmarks that include a variety of continuous robot locomotion control tasks commonly used in previous works (Chow et al., 2019; Zheng et al., 2023; Liu et al., 2023b) and `DSRL` (Liu et al., 2023a), a comprehensive benchmark specialized for offline safe RL, provides the offline datasets. We consider three tasks: `Run`, `Circle`, and `Velocity` and multiple types of robots: `Ant`, `Ball`, `Car`, `Drone`, `Hopper`, and `HalfCheetah`. In the `Run` task, agents earn rewards for achieving high speeds between two boundaries but face penalties if they cross the boundaries or exceed an agent-specific velocity threshold. In the `Velocity` task, agents also receive rewards for

| Taks | Metric | BC-safe | CQL-Saute | BCQ-Lag | BEAR-Lag | CPQ | COptiDICE | VOCE | CDT | TREBI | FISOR | Ours |
|---|---|---|---|---|---|---|---|---|---|---|---|---|
| Ball-Run | Reward ↑ | 0.36±0.32 | 1.95±0.94 | **0.5±0.1** | 2.07±0.75 | 0.89±0.15 | 1.76±0.24 | 2.61±1.27 | **0.91±0.09** | 0.78±0.23 | **0.76±0.06** | **0.97±0.01** |
| | Cost ↓ | 2.08±3.63 | 9.26±6.33 | **0.35±1.15** | 16.23±1.72 | 2.09±2.96 | 11.59±1.58 | 12.74±7.58 | **0.92±0.31** | 2.19±4.14 | **0.0±0.0** | **0.27±0.19** |
| Car-Run | Reward ↑ | 0.96±0.03 | **0.93±0.04** | 0.84±0.01 | 1.0±0.01 | **0.97±0.01** | 0.95±0.03 | 0.98±0.01 | **1.0±0.0** | 0.97±0.03 | 0.73±0.18 | **0.95±0.04** |
| | Cost ↓ | 1.06±2.04 | **0.67±1.24** | 2.63±1.48 | 6.47±5.87 | **0.0±0.0** | **0.0±0.0** | 9.41±5.82 | **0.91±0.72** | 2.34±1.47 | 2.37±2.96 | **0.19±0.27** |
| Ant-Circle | Reward ↑ | 0.56±0.33 | **0.2±0.11** | 1.21±0.34 | 1.09±0.38 | **0.04±0.1** | 0.35±0.26 | **-0.0±0.0** | 0.82±0.29 | 0.11±0.13 | **0.16±0.13** | **1.01±0.26** |
| | Cost ↓ | 3.5±5.73 | **0.0±0.0** | 19.09±7.64 | 17.06±9.64 | **0.0±0.0** | 15.55±21.06 | **0.0±0.0** | 6.75±4.71 | 7.95±12.99 | **0.0±0.0** | **0.55±1.57** |
| Ball-Circle | Reward ↑ | 0.62±0.19 | **0.65±0.13** | 0.93±0.13 | 1.08±0.15 | 0.93±0.04 | 0.92±0.08 | 0.0±0.01 | 0.91±0.05 | 0.73±0.1 | **0.26±0.14** | **0.87±0.03** |
| | Cost ↓ | 1.67±1.52 | **0.02±0.18** | 8.95±2.64 | 11.22±1.67 | 2.25±3.18 | 9.1±1.85 | 12.96±5.12 | 2.11±1.0 | 1.8±1.19 | **0.0±0.0** | **0.0± 0.0** |
| Car-Circle | Reward ↑ | 0.38±0.27 | **0.54±0.26** | 0.69±0.32 | 1.06±0.08 | 0.69±0.38 | 0.59±0.1 | 0.19±0.1 | **0.88±0.05** | 0.52±0.13 | **0.25±0.16** | **0.85±0.04** |
| | Cost ↓ | **0.58±1.46** | 2.84±3.8 | 11.74±7.67 | 15.04±2.79 | 7.57±9.32 | 10.95±5.69 | 1.56±2.59 | **1.59±1.52** | 1.12±2.92 | **0.0±0.0** | **0.73±1.95** |
| Drone-Circle | Reward ↑ | 0.82±0.16 | 0.15±0.1 | 1.38±0.03 | 1.27±0.05 | 0.83±0.1 | 0.59±0.05 | **0.0±0.0** | 0.94±0.03 | 0.68±0.14 | **0.62±0.12** | **0.82±0.11** |
| | Cost ↓ | 1.32±1.49 | 2.49±7.06 | 19.07±1.2 | 14.88±1.84 | 3.66±4.64 | 2.62±2.13 | **0.0±0.0** | 1.44±0.85 | 5.89±3.93 | **0.06±0.34** | **0.07±0.54** |
| Ant-Velocity | Reward ↑ | 0.98±0.06 | -0.09±0.33 | 0.93±0.22 | -1.06±0.0 | -0.63±0.5 | 1.05±0.01 | -1.02±0.0 | **1.02±0.02** | 0.59±0.18 | 0.78±0.04 | **0.9±0.05** |
| | Cost ↓ | 0.53±0.35 | 1.0±0.53 | 15.94±11.08 | **0.0±0.0** | 0.9±1.55 | 11.09±2.85 | **0.0±0.0** | **0.59±0.32** | 0.9±0.48 | **0.0± 0.0** | **0.58±0.15** |
| HalfCheetah-Velocity | Reward ↑ | **0.9±0.04** | 0.54±0.54 | 1.08±0.04 | 1.06±0.01 | 1.79±0.02 | **0.69±0.05** | -0.01±0.18 | 1.01±0.01 | **0.46±0.06** | **0.86 ± 0.02** | **0.96±0.04** |
| | Cost ↓ | **0.6±0.63** | 1.28±0.42 | 59.17±23.48 | 29.82±15.74 | 97.81±0.21 | **0.0±0.0** | **0.0±0.0** | 1.18±0.69 | **0.05±0.13** | **0.0 ± 0.0** | **0.79±0.2** |
| Hopper-Velocity | Reward ↑ | **0.25±0.26** | 0.09±0.11 | 0.51±0.3 | 0.5±0.02 | 0.39±0.27 | 0.15±0.17 | 0.15±0.21 | 0.58±0.38 | 0.61±0.3 | **0.16 ± 0.07** | **0.89±0.02** |
| | Cost ↓ | **0.6±0.87** | 3.02±3.05 | 13.53±5.64 | 18.91±1.12 | 15.31±14.65 | 5.99±5.97 | 2.07±3.1 | 1.05±0.77 | 8.52±9.58 | **0.0 ± 0.0** | **0.32±0.23** |
| Average | Reward ↑ | 0.66±0.35 | 0.55±0.7 | 0.89±0.35 | 0.9±0.84 | 0.66±0.68 | 0.78±0.46 | 0.32±1.03 | 0.9±0.21 | 0.6±0.28 | **0.51±0.3** | **0.91±0.11** |
| | Cost ↓ | 1.14±2.39 | 2.29±4.48 | 16.72±18.89 | 14.4±10.23 | 14.4±30.48 | 7.43±9.29 | 4.31±6.61 | 1.84±2.51 | 3.42±6.55 | **0.27±1.24** | **0.39±0.94** |

Table 1: Evaluation results of the normalized reward and cost. The agents are trained using the whole dataset. The cost threshold is 1 (the cost budget is 5 for `Run` and `Circle` tasks and 10 for `Velocity` task). The percentages of safe trajectories in the datasets are as follows: $4.9\%, 28.6\%,$ $5.9\%, 5.5\%, 6.8\%, 9.1\%, 9.3\%, 9.7\%, 5.8\%$ for the tasks listed from top to bottom. ↑: the higher the reward, the better. ↓: the lower the cost (up to 1), the better. Each value is averaged over 20 episodes and 3 seeds. **Bold**: Safe agents. Gray: Unsafe agents. **Blue**: Safe agent with the highest reward.

moving forward but incur costs if they exceed a specified velocity threshold. In the `Circle` task, agents are rewarded for moving in a clockwise circular pattern but are constrained within a safe region smaller than the radius of the target circle. This setup of conflicting rewards and costs requires the agents to balance their trade-off (Liu et al., 2022; Guo et al., 2024), which is challenging since pursuing high rewards can lead to constraint violations, while conservative policies yield safety but lower rewards. More details of the environments can be found in Appendix B.

**Metrics.** Our evaluation metrics include normalized cumulative reward and cost:

$$R_{\text{normalized}} = \frac{R_\pi - R_{\min,\kappa}}{R_{\max,\kappa} - R_{\min,\kappa}} \qquad C_{\text{normalized}} = \frac{C_\pi}{\kappa}$$

where $R_\pi$ and $C_\pi$ are the evaluated cumulative reward and cost of a policy $\pi$, respectively, and $R_{\max,\kappa}$ and $R_{\min,\kappa}$ are the maximum and minimum cumulative reward of the trajectories that satisfy the cost budget $\kappa$ in the offline dataset $\mathcal{D}$. For convenience, we will abbreviate normalized cumulative reward as reward and cumulative cost as cost. A policy is safe if $C_{\text{normalized}} \leq 1$.

**Baselines**. We compare CCAC with the following baselines: 1) BC-safe: behavior cloning that imitates only safe trajectories in the datasets. 2) CQL-Sauté: a state-augment-based method that we adapt from Sauté (Sootla et al., 2022) and CQL (Kumar et al., 2020). 3, 4) BCQ-Lagrangian (BCQ-Lag) and BEAR-Lagrangian (BEAR-Lag): Lagrangian-based methods built upon BCQ (Fujimoto et al., 2018) and BEAR (Kumar et al., 2019) respectively. 5) CPQ (Xu et al., 2022): a $Q$-learning-based method that treats OOD actions as unsafe and learns policy from safe actions. 6) CoptiDICE (Lee et al., 2022): a stationary distribution correction based method. 7) VOCE (Guan et al., 2024): a probabilistic-inference-based method that learns conservative critics. 8) CDT (Liu et al., 2023b): a Decision-Transformer based method that learns conditioned policies. 9) FISOR (Zheng et al., 2023): a diffusion-based method that learns a feasibility-guided policy. 10) TREBI (Lin et al., 2023): a diffusion-based method that uses trajectory optimization. See Appendix C for implementation details. For CQL-Sauté, CDT, TREBI, and our method, an initial cost budget is assigned for each rollout and then updated based on the incurred costs. For the rest of the baselines, we use the cost budget to refer to the constant cost threshold used for training and evaluation.

## 5.1 CAN CCAC LEARN SAFE AND HIGH-REWARD POLICIES FROM OFFLINE DATASETS?

The evaluation results for different trained policies are presented in Table 1. The training curves are included in Appendix D.1. *Our method demonstrates the best performance compared to the*

*baselines in terms of reaching the highest reward in most of the tasks while maintaining safety.* We can also observe that high rewards usually come with high costs and vice versa, which highlights their trade-off. The baseline methods either suffer from significant constraint violations or yield sub-optimal returns. The results of BC-safe and CQL-Sauté indicate that, due to distribution shifts, merely mimicking the safe data or maximizing the reward critic of cost-augmented states cannot achieve safety. The Lagrangian-based baselines fail to behave safely on most tasks, which suggests that directly applying the primal-dual method in the offline setting can hardly work well without regularizing the OOD state-action pairs. Though CPQ considers the OOD actions during training, the OOD states at test time can still result in constraint violations. COptiDICE and VOCE oscillate between overly conservative and overly aggressive since the trajectories in the dataset are collected from diverse behavior policies, potentially leading to biased estimations of the critics. FISOR exhibits conservative behaviors as it considers hard constraint instead of soft constraint. TREBI struggles to maintain safety as it restricts its learned policy to remain close to the behavior policy, despite the behavior policy being suboptimal.

**Distribution shifts.** To assess the effect of OOD states and actions, we use different percentages of data to train policies and then evaluate their performance. Less training data makes it more challenging to train the policies and implies a larger distribution shift during evaluation. The evaluation results are shown in Figure 2. Appendix B.2 also includes a visualization of the offline datasets. We can observe that most methods exhibit an increase in costs and a decrease in rewards as the percentage of training data decreases. Our method, on the other hand, continues to satisfy the constraints and achieve high rewards, showcasing its robustness against distribution shifts.

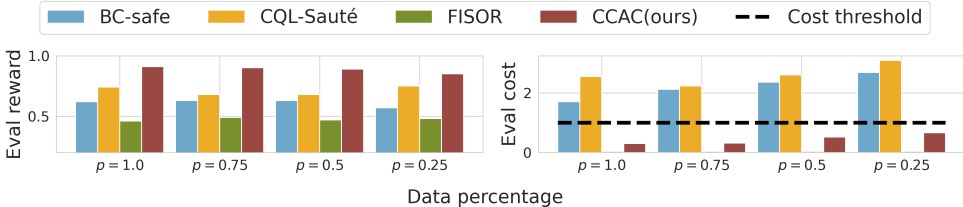

Figure 2: Evaluation results of reward and cost in `Run` and `Circle` tasks with different percentages of datasets being used for training. The dashed line represents the normalized cost threshold.

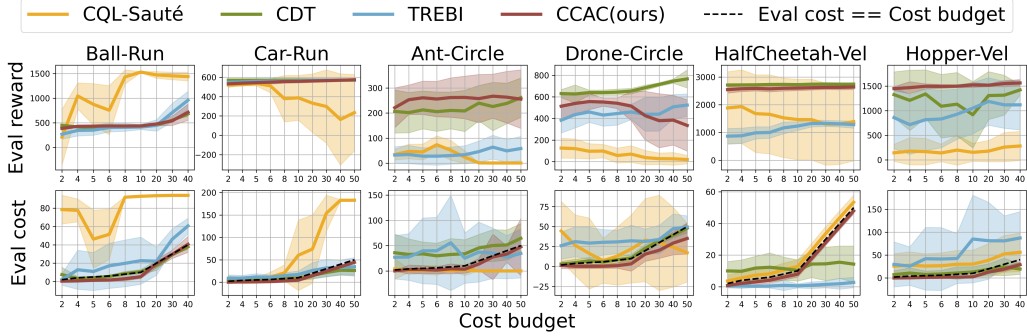

Figure 3: Results of alignment with different cost budgets. The agents are trained using the whole dataset. The top-row plots show the evaluated reward and the bottom-row plots show the evaluated cost. The solid line and the light shade area represent the mean and mean $\pm$ standard deviation. The dashed line represents the scenario where the evaluated cost matches the cost budget.

## 5.2 CAN CCAC ACHIEVE ZERO-SHOT ADAPTION TO DIFFERENT COST BUDGETS?

One major advantage of our method is the ability to adapt to varying cost budgets without the need for re-training. To illustrate this, we set different cost budgets for evaluation rollouts to obtain the results in Figure 3. We compare our method with CQL-Sauté, CDT, and TREBI, since all other baselines cannot adapt to different cost budgets without re-training. *The results reveal a strong correlation between the actual and target cost for our method.* We can also observe that the actual

cost of our method is consistently below the cost budget even when the budget is small, whereas CDT fails to ensure safety despite achieving higher rewards in some cases. Moreover, our method achieves better zero-shot adaptation by optimizing towards safe actions using constraint-conditioned critics and actor built with simple MLP models, unlike CDT and TREBI, which relies on the sequential modeling abilities of transformer architectures and planning capacities of the diffusion models, respectively, to mimic the trajectories in the dataset.

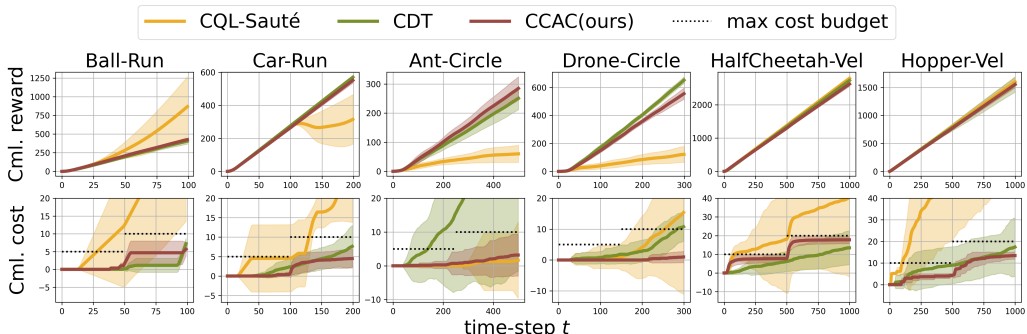

Figure 4: Results of cumulative reward and cost over time. The cost budget is 5 (increased to 10 at $t = T/2$) for `Run` and `Circle` tasks and 10 (increased to 20 at $t = T/2$) for `velocity` tasks.

To further evaluate the adaptability to manage unseen cost budgets, we construct two more challenging scenarios: (1) we manually assign an increased cost budget at a certain time-step during evaluation and examine if the agents can respond to the change accordingly. Note that all the training data encounters a gradually decreasing cost budget as the cost budget is updated as $\kappa_{t+1} = \kappa_t - c_t$. Figure 4 shows the cumulative reward and cost over time. We can observe that our method can adapt to changes in the cost budget without compromising rewards or safety satisfaction, while other baselines either violate constraints or suffer a notable drop in rewards. (2) We build a modified dataset by removing all the safe trajectories whose cumulative cost is smaller than a certain threshold. We re-train the agents and evaluate them using an unseen cost budget that is set to be half of this threshold. The average evaluation results are presented in Table 2, with the detailed results for each task provided in Appendix D.3. Additionally, for CQL-Sauté, CDT, and our method, we evaluate different additional unseen cost budgets that are all smaller than the threshold, and the adaptation results are shown in Figure 10. The results show that our method consistently attains high rewards while maintaining safety, whereas all baselines either fail to ensure safety or exhibit overly conservative behavior, highlighting our method's better adaptability and robustness to distribution shifts. On the other hand, the adaptability to unseen cost budgets is also linked to the so-called "stitching" ability (Kumar et al., 2022). Despite the removal of safe trajectories, certain sub-trajectories within the unsafe ones can still be safe. The results suggest that our method is able to exploit these better than the existing techniques.

| Tasks | Metric | CQL-Sauté | BCQ-Lag | BEAR-Lag | CPQ | COptiDICE | VOCE | CDT | TREBI | FISOR | CCAC(ours) |
|---|---|---|---|---|---|---|---|---|---|---|---|
| Run | Reward ↑ | 0.55±0.44 | 1.51±0.91 | 1.88±1.03 | 0.95±0.03 | 1.38±0.5 | 1.65±1.07 | 0.99±0.02 | 0.82±0.19 | **0.74±0.08** | **0.96±0.03** |
| | Cost ↓ | 5.23±6.26 | 19.85±12.76 | 23.56±7.99 | 1.92±2.66 | 6.27±6.29 | 9.35±8.95 | 1.34±0.74 | 2.12±2.36 | **0.54±1.90** | **0.23±0.27** |
| Circle | Reward ↑ | 0.4±0.33 | 1.11±0.31 | 1.13±0.21 | 0.64±0.42 | 0.63±0.25 | 0.06±0.11 | 0.91±0.17 | 0.43±0.25 | **0.27±0.18** | **0.79±0.24** |
| | Cost ↓ | 6.88±9.15 | 15.61±6.1 | 13.19±6.18 | 3.73±6.43 | 9.82±10.57 | 10.05±15.53 | 2.58±2.49 | 3.53±8.26 | **0.17±0.76** | **0.17±0.79** |
| Velocity | Reward ↑ | 0.44±0.43 | 0.78±0.37 | 0.18±0.8 | 0.41±1.15 | 0.65±0.36 | -0.41±0.44 | 0.88±0.24 | 0.45±0.24 | **0.47±0.26** | **0.86±0.2** |
| | Cost ↓ | 2.08±3.23 | 27.26±29.75 | 20.1±25.76 | 35.18±44.66 | 6.94±7.22 | **0.0±0.0** | 2.91±2.87 | 1.07±2.52 | **0.08±0.27** | **0.38±0.2** |
| Average | Reward ↑ | 0.45±0.4 | 1.09±0.58 | 0.98±0.93 | 0.63±0.75 | 0.8±0.47 | 0.26±0.96 | 0.91±0.19 | 0.51±0.28 | **0.44±0.27** | **0.85±0.21** |
| | Cost ↓ | 4.91±7.33 | 20.44±19.33 | 17.8±16.46 | 13.81±30.22 | 8.07±8.85 | 6.54±12.1 | 2.41±2.45 | 2.41±6.0 | **0.22±1.05** | **0.25±0.56** |

Table 2: Average evaluation results of the normalized reward and cost of agents trained on datasets with safe trajectories removed. The cost threshold is 1 (the cost budget is 5 for `Run` and `Circle` tasks and 10 for `Velocity` tasks). Each value is averaged over 20 episodes and 3 seeds. **Bold**: Safe agents. Gray: Unsafe agents. **Blue**: Safe agent with the highest reward.

## 5.3 WHAT IS THE IMPORTANCE OF THE OOD DETECTION COMPONENT IN CCAC?

To assess the importance of the OOD detection part in our method, i.e., the CVAE and the classifier, we first conduct an ablation study by constructing three variants of our method as shown in Figure 5, where "w/ CVAE" and "w/ offline data" mean that the OOD data used to train the cost critic are either sampled from the CVAE and identified by the classifier, or obtained from the offline datasets based-on cost budgets; "w/o CVAE" and "w/o offline data" mean the absence of these elements. Note that CVAE(w/ CVAE & w/o offline data) is the CVAE(ours) in the remaining tables and figures. We can observe that when using both the generated OOD data and OOD offline data, the performance does not change much since the OOD distribution $\nu(s, a|\kappa)$ that the CVAE aims to approximate covers the OOD offline data. The result also shows that the limited OOD offline data only offers limited benefits in learning the critics and policies. The CVAE and classifier are essential components as their removal leads to substantial increases in cost and moderate drops in rewards. The CVAE enables the generation of diverse data not present in the offline dataset, and the classifier can determine which of these generated samples are OOD.

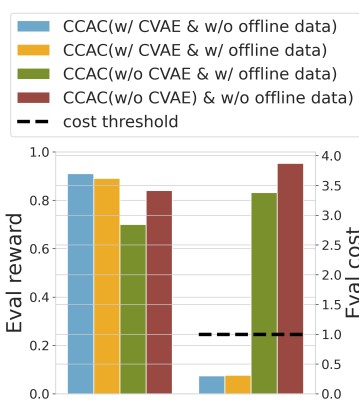

Figure 5: Ablation study: average performance of CCAC and its variants in `Run` and `Circle` tasks.

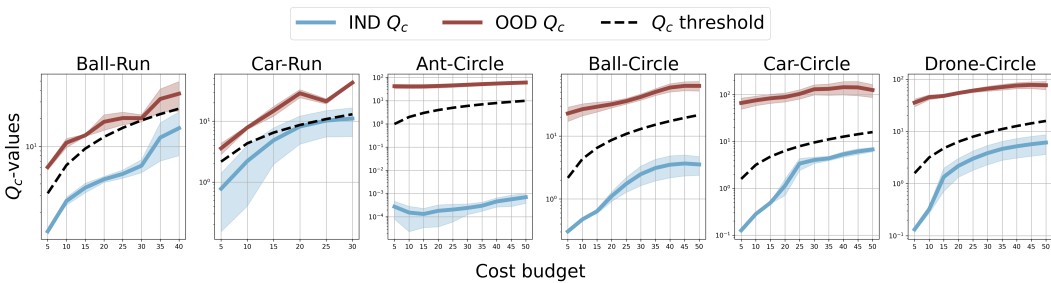

Figure 6: Ablation study: $Q_c$-values plots. The solid line and the light shade area represent the mean and mean $\pm$ standard deviation, respectively. The dashed line denotes the $Q_c$-value thresholds.

We also evaluate the $Q_c$-values for both the IND and OOD state-action pairs. For a cost budget $\kappa$, we sample state-action pairs whose cost-to-go satisfies the threshold from the offline dataset and compute $E_{s,a\sim\mathcal{D}}[Q_c(s, a|\kappa)]$ as IND $Q_c$. Next, we generate state-action pairs using the CVAE, identify the OOD samples using the classifier, and compute $E_{s,a\sim\nu}[Q_c(s, a|\kappa)]$ as OOD $Q_c$. The results are shown in Figure 6. We can observe that the IND $Q_c$ satisfies the threshold while OOD $Q_c$ exceeds the threshold. The results are consistent with Eq.(5) which aims to overestimate the $Q_c$-values of OOD state-action pairs while minimizing the $Q_c$-values of IND state-action pairs. This indicates the effectiveness of the data generation and detection components in our method.

## 6  CONCLUSION

We propose a novel method called CCAC that leverages the constraint information to guide policy learning and deployment with the support of constraint-related data generation and OOD detection. Empirical results demonstrate that CCAC excels in learning a safe and high-reward policy across multiple offline safe RL tasks, whereas prior works either show overly conservative behaviors or struggle to meet safety requirements. Notably, CCAC can adapt to different or even unseen cost budgets without re-training and consistently maintain safety and achieve high rewards. The ablation study highlights the critical role of data generation and OOD detection in our method. A bottleneck of our method could be the data generation and OOD detection component when the dimension of state-action increases. However, we show a feasible direction by integrating such techniques to regularize critics and policy training. Future works will investigate the incorporation of more advanced data generation and OOD detection techniques and the possibility of offline-to-online finetuning to further improve policy performance since the offline dataset can be sub-optimal.

ACKNOWLEDGEMENT

The authors thank the anonymous reviewers for their invaluable feedback and constructive suggestions. This work was supported in part by the U.S. National Science Foundation under grant CCF-2340776.

ETHICS STATEMENT

This work does not present any significant ethical concerns. All experiments were carried out in simulated environments, preventing any risk to real-world systems. The focus on offline datasets ensures the avoidance of potentially dangerous and unsafe interactions during policy learning, promoting safety and responsible practices in reinforcement learning.

REPRODUCIBILITY STATEMENT

The datasets used are provided from a publicly available benchmark that uses simulated dynamical control environments, ensuring that the experimental setup can be reproduced. Details about the experimental setup, including parameters, models, evaluation metrics, and implementations, are provided in the appendix and supplementary materials to facilitate the reproduction of our findings.

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

## A    PROOF OF THEOREM 4.1

*Proof.* Let $Q_c^k$ denote the true $Q$-function at iteration $k$ and $\hat{Q}_c^k$ denote the $k$-th $Q$-function obtained by solving Eq.(5). $Q_c^{k+1}$ is related to the previous $Q_c^k$: $Q_c^{k+1} = \mathcal{T}^\pi Q_c^k$. By introducing the Lagrangian multiplier $\lambda_c$, Eq.(5) becomes Eq.(6), then take the derivative of Eq.(6) w.r.t. $Q_c$ and set it to 0, we obtain the following expression for $\hat{Q}_c^{k+1}$ in terms of in terms of $\hat{Q}_c^k$:

$$\hat{Q}_c^{k+1}(s,a|\kappa) = \mathcal{T}\hat{Q}_c^k(s,a|\kappa) + \frac{\lambda_c}{2} \cdot \frac{\nu(s,a|\kappa)}{\pi_\mathcal{D}(s,a|\kappa)} \, , \forall k \tag{10}$$

Note that we only sample $\kappa$ once from $\mathcal{D}$, so the $\kappa \sim \mathcal{D}$ in the objective and constraint in Eq.(5) are the same and will cancel out in Eq.(10). Since $\nu(s,a|\kappa) \geq 0$, $\lambda_c \geq 0$, $\pi_\mathcal{D}(s,a|\kappa) > 0$, at each iteration $\hat{Q}_c^{k+1}(s,a|\kappa) \geq \hat{Q}_c^k(s,a|\kappa)$. Now, let's examine the fixed point of Eq.(10):

$$\hat{Q}_c^\pi(s,a|\kappa) = \mathcal{T}\hat{Q}_c^\pi(s,a|\kappa) + \frac{\lambda_c}{2} \cdot \frac{\nu(s,a|\kappa)}{\pi_\mathcal{D}(s,a|\kappa)}$$

$$= c + \gamma P^\pi \hat{Q}_c^\pi(s,a|\kappa) + \frac{\lambda_c}{2} \cdot \frac{\nu(s,a|\kappa)}{\pi_\mathcal{D}(s,a|\kappa)}$$

$$= Q_c^\pi(s,a|\kappa)(I - \gamma P^\pi) + \gamma P^\pi \hat{Q}_c^\pi(s,a|\kappa) + \frac{\lambda_c}{2} \cdot \frac{\nu(s,a|\kappa)}{\pi_\mathcal{D}(s,a|\kappa)}$$

$$= Q_c^\pi(s,a|\kappa) + \gamma P^\pi[\hat{Q}_c^\pi(s,a|\kappa) - Q_c^\pi(s,a|\kappa)] + \frac{\lambda_c}{2} \cdot \frac{\nu(s,a|\kappa)}{\pi_\mathcal{D}(s,a|\kappa)}$$

$$\Rightarrow \hat{Q}_c^\pi(s,a|\kappa) = Q_c^\pi(s,a|\kappa) + \frac{\lambda_c}{2}(I - \gamma P^\pi)^{-1} \frac{\nu(s,a|\kappa)}{\pi_\mathcal{D}(s,a|\kappa)}$$

We set $\lambda_c \geq 0$ since we don't want to underestimate the $Q_c$-values. When $Q_c^\pi(s,a|\kappa) > \kappa$, it means that the state-action pairs are OOD, e.g., unsafe to take the actions in the states, then we want to overestimate $\hat{Q}_c^\pi(s,a|\kappa)$, the choice of $\lambda_c$ that guarantee $\hat{Q}_c^\pi(s,a|\kappa) \geq \epsilon$ is given by:

$$\lambda_c \geq 2(\epsilon - Q_c^\pi(s,a|\kappa))(I - \gamma P^\pi)\Big[\frac{\pi_\mathcal{D}(s,a|\kappa)}{\nu(s,a|\kappa)}\Big]$$

$$\Rightarrow \lambda_c \geq 2\max_{s,a,\kappa}(\epsilon - Q_c^\pi(s,a|\kappa))(I - \gamma P^\pi)\Big[\frac{\pi_\mathcal{D}(s,a|\kappa)}{\nu(s,a|\kappa)}\Big]$$

We show that the $Q_c$-values of OOD state-action pairs can be overestimated to exceed the threshold.

## B    ENVIRONMENTS DETAILS

### B.1    REWARD AND COST FUNCTIONS

We use the `Bullet-safety-gym` (Gronauer, 2022) and `Safety-Gymnasium` (Ji et al., 2023) for this set of experiments. We consider three tasks: `Run`, `Circle`, and `Velocity` and multiple types of robots: `Ant`, `Ball`, `Car`, `Drone`, `Hopper`, and `HalfCheetah`.

In the `Run` task, agents receive rewards for high-speed movement between two safety boundaries. However, they incur penalties when they either cross these boundaries or surpass a velocity threshold that is specific to different types of robots. In the `Velocity` task, agents also receive rewards for moving forward but incur costs if they exceed a specified velocity threshold. The reward and cost function are defined as:

$$r(\mathbf{s_t}) = ||\mathbf{x_{t-1}} - \mathbf{g}||_2 - ||\mathbf{x}_t - \mathbf{g}||_2$$
$$c(\mathbf{s_t}) = \mathbf{1}(|y| > y_{\text{lim}} \text{ or } ||\mathbf{v_t}||_2 > v_{\text{lim}})$$

where $s_t = [x_t, y_t, v_x, v_y]$, $r$ is the radius of the circle, and $x_{\text{lim}}$ specifies the range of the safety region. In the `Circle` task, agents gain rewards for circular motion in a clockwise direction but are required to remain inside a designated safe area, which is smaller than the circumference of the intended circle. The reward and cost functions are defined as:

$$r(\mathbf{s_t}) = \frac{-y_t v_x + x_t v_y}{1 + ||\sqrt{x_t^2 + y_t^2} - r|}$$
$$c(\mathbf{s_t}) = \mathbf{1}(|x| > x_{\text{lim}})$$

where $y_{\lim}$ is the safety boundary and $v_{\lim}$ is the velocity limit. Since the cost function is the indicator function so the cost-to-go represents the number of constraint violations.

## B.2 OFFLINE DATASET VISUALIZATION

The dataset cost-reward plots for tasks `Ball-Run`, `Car-Run`, `Ant-Circle`, `Ball-Circle`, `Car-Circle`, and `Drone-Circle`, are shown in Figure 7. Analyzing the figures provided, we can generally discern an increasing trend for the reward in relation to the cost. In other words, as cost increases, so too might the reward return, underscoring the inherent trade-off between reward and cost. This phenomenon aligns with findings discussed in previous works (Liu et al., 2023b; 2022; Guo et al., 2024). We utilize the data density filter in Liu et al. (2023a) to evaluate the algorithms' ability to perform under different levels of data availability, e.g., less training data means a larger distribution shift during evaluation, and assess their generalization capabilities. We also use the partial data filter to filter out some specific trajectories, as shown in Figure 8. For `Run` and `Circle` tasks, the safe trajectories whose cumulative cost is smaller than 10 are removed. For `Velocity` tasks, the safe trajectories whose cumulative cost is smaller than 20 are removed.

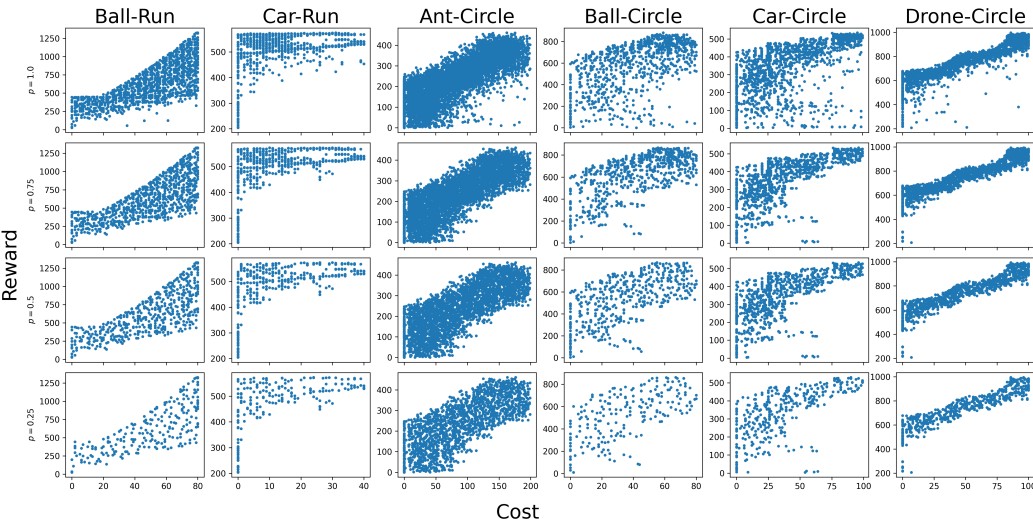

Figure 7: Illustration of the offline dataset. Each row shows reward versus cost for varying data percentages. Each column represents an environment. Each point denotes a collected trajectory with corresponding episodic reward and cost values.

## C IMPLEMENTATION DETAILS

### C.1 OUR METHOD

We present the full algorithm of our method in Algorithm 1. There are two sets of components that need training: the first includes the RL components, i.e., the critics and the policies; the second includes the data generation components, i.e., the CVAE and the classifier. Directly training all these components simultaneously would make it difficult to converge since, at the beginning of training, the poor-quality OOD data generated by the CVAE and a weak classifier that cannot correctly identify OOD samples would lead to an erroneous update of the critics and policies, especially the cost critic. Therefore, we train these two sets of components separately. We first train the CVAE and classifier for $M$ iterations until convergence. Then we perform policy training which follows the standard training procedure for deep RL methods: at each iteration, we sample a batch of data, update critics, update the actor, update the target critics, and so on. To create OOD samples, we sample randomly in the latent space, $z$, and use the CVAE decoder to produce state-action pairs. These generated pairs are then fed into the classifier, which outputs the probability, $h(s, a|\kappa)$, of being OOD. We classify the pair as OOD if $h(s, a|\kappa)$ exceeds a threshold of 0.5. For evaluation, we first specify an initial cost budget $\kappa_1$, and then this cost budget is updated autoregressively based on constraint

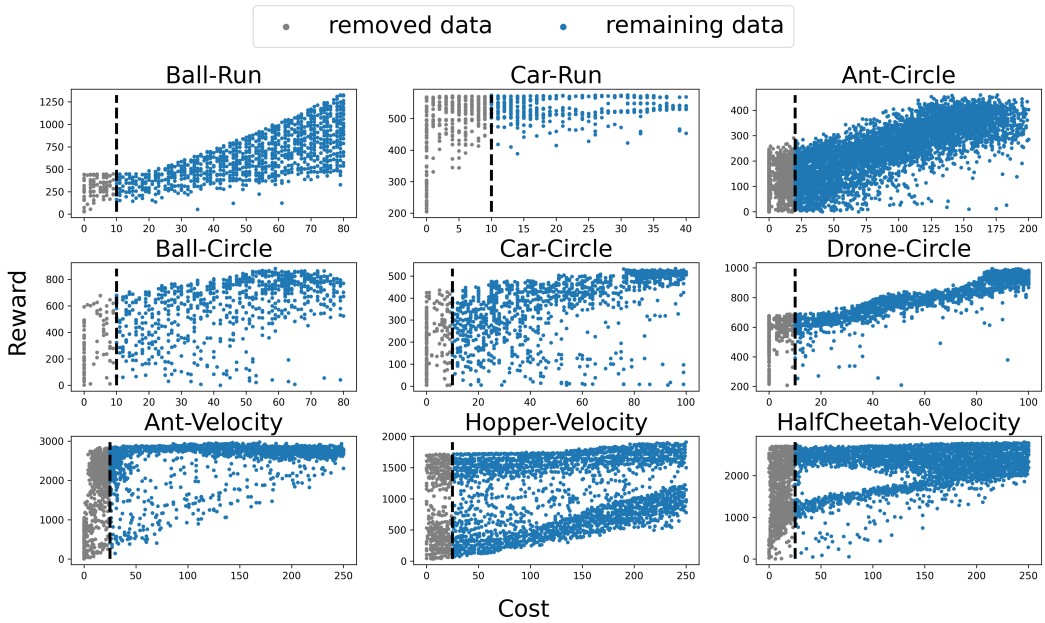

Figure 8: Illustration of the offline dataset with partial data removed. Each point denotes a collected trajectory with corresponding episodic reward and cost values.

violations $\kappa_{t+1} = \kappa_t - c_t$. Then, we compute the cumulative cost $\sum_{t=1}^{T} c_t$ and compare it with the initial cost budget. A smaller cumulative cost means safety is achieved.

---

**Algorithm 1** Cost-Conditioned Actor-Critic (CCAC)

---

**Require:** offline data $\mathcal{D}$
1:  // CVAE and Classifier Training
2:  Initialize encoder $p_\varphi$, decoder $q_\rho$, and classifier $h_\psi$
3:  **for** $t = 0, ..., M$ **do**
4:      Sample a batch of data $(s, a, \kappa) \sim \mathcal{D}$ and a cost threshold $\bar{\kappa}$
5:      Update encoder $p_\varphi$ and decoder $q_\rho$ by Eq.(3)
6:      Get reconstructed state-action pairs $(\hat{s}, \hat{a})$ via CVAE
7:      Assign the labels $y$ for each state-action pair based on $\kappa$ and $\bar{\kappa}$
8:      Update classifier $h_\psi$ by Eq.(4)
9:  **end for**
10: // Policy Training
11: Initialize reward critic ensemble $\{Q_{r_i}(s, a|\kappa)\}_{i=1}^{n}$ and cost critic ensemble $\{Q_{c_i}(s, a|\kappa)\}_{i=1}^{n}$, actor $\pi_\theta$, Lagrangian multiplier $\lambda_c$ and $\lambda_a(\kappa)$, target networks $\{Q'_{r_i}\}_{i=1}^{n}$ and $\{Q'_{c_i}\}_{i=1}^{n}$ with $\phi'_{r_i} \leftarrow \phi'_{r_i}$ and $\phi'_{c_i} \leftarrow \phi'_{c_i}$
12: **for** $t = 0, ..., N$ **do**
13:     Sample a batch as data $(s, a, r, c, \kappa, s') \sim \mathcal{D}$
14:     (Optional) Perform one iteration of CVAE and classifier training
15:     Sample the same size of $z \sim p(z|\kappa)$ conditioned on the $\kappa$ in the batch
16:     Generate state-action pairs $(\hat{s}, \hat{a}) \sim p_\varphi(s, a|z, \kappa)$
17:     Assign labels to $(\hat{s}, \hat{a})$ using the classifier
18:     Update cost critics and $\lambda_c$ by Eq.(6) and reward critics by Eq.(7)
19:     Update actor and $\lambda_a(\kappa)$ by Eq.(9)
20:     Update target cost critic: $\phi'_c \leftarrow \alpha \phi_c + (1 - \alpha)\phi'_c$
21:     Update target reward critic: $\phi'_r \leftarrow \alpha \phi_r + (1 - \alpha)\phi'_r$
22: **end for**

---

### C.2 HYPERPARAMETERS

For BCQ-Lag, BEAR-Lag, CPQ, COptiDICE and CDT, we use the `OSRL`[1] implementation and we implement our method in this framework. We adopt the CQL-Sauté from this CQL implementation[2]. For VOCE[3], FISOR[4], and TREBI[5], we use their official implementations. For value-based baselines, we use Gaussian policies with mean vectors given as the outputs of neural networks, and with variances that are separate learnable parameters. The policy networks and Q networks for all experiments have two hidden layers with ReLU activation functions. The $K_P$, $K_I$ and $K_D$ are the PID parameters Stooke et al. (2020) that control the Lagrangian multiplier for the BCQ-Lag and BEAR-Lag. For each task, the update steps is the same for all the methods. For fair comparisons, we use the same model structure for the critics and policies for all the value-based methods. However, CDT and FISOR take advantage of more expressive transformers and diffusion models, making it difficult to make fair comparisons: CDT uses a Transformer model to take a sequence of states and actions as input, whereas our method only takes in current state; FISOR uses a ResNet-based diffusion model as its policy network, whereas ours is a simple MLP. When switching from the diffusion model to an MLP architecture, FISOR results in significant performance degradation. Therefore, we report results based on their original design and default hyperparameters. BCQ-Lag and BEAR-Lag employ VAE to generate actions, while our method uses CVAE to generate state-action pairs. The constraint threshold is 5 across all the tasks. The hyperparameters that are not mentioned are in their default value for baselines. The hyperparameters used in the experiments are shown in Table 3.

| Parameter | Ball-Run | Car-Run | Ant-Circle | Ball-Circle | Car-Circle | Drone-Circle | Velocity |
|---|---|---|---|---|---|---|---|
| Actor hidden size | | | | [256, 256] | | | |
| Critic hidden size | | | | [256, 256] | | | |
| VAE/CVAE hidden size | | | | [512, 512, 64, 512, 512] | | | |
| Episode length | 100 | 200 | 500 | 200 | 300 | 300 | 1000 |
| $[K_P, K_I, K_D]$ | | | [0.1, 0.003, 0.001] BCQ-Lag, BEAR-Lag | | | | |
| Batch size | 512 | 1024 | 512 | 512 | 512 | 512 | 2048 |
| Training steps | 2e5 | 1e5 | 2e5 | 2e5 | 2e5 | 3e5 | 2e5 |
| $\gamma$ | | | | 0.99 | | | |
| Actor learning rate | | | | 1e-4 | | | |
| Critic learning rate | | | | 1e-3 | | | |
| VAE/CVAE learning rate | | | | 1e-3 | | | |
| Critic ensemble | | | | 4 | | | |

Table 3: Hyperparameters for value-based methods (including our method).

## D MORE EXPERIMENT RESULTS

### D.1 TRAINING CURVES

The training curves of our method and the baseline methods are presented in Figure 9. We evaluate the methods at regular training intervals, which is optional, and the interaction data is not added to the offline dataset used to train the methods. These curves follow the trends observed in online model-free safe RL methods, where costs typically start high and decrease over time, while rewards begin low and increase. In the offline setting, data collection is not needed and training safety is ensured due to the offline datasets. After convergence, our method consistently remains below the cost threshold and achieves high rewards, while other baselines either struggle to satisfy constraints or suffer from conservatism. Note that for our method, the constraint threshold is only used during evaluation; during training, the method is unaware of the threshold under which it will be evaluated. As mentioned in section 5, we consider the scenarios where pursuing rewards aggressively will lead to significant costs and excessive demand for safety will lead to low rewards. In our experiments, for `Circle` task, traveling along the circle will get high rewards but will enter unsafe regions; for `Run` task, driving as far as possible will get high rewards but will exceed the speed limits. On the other hand, the agents can simply take no actions and stand still to achieve safety. The training curves reveal such a trade-off between rewards and costs. For example, BCQ-Lag, BEAR-Lag, and

---

[1]https://github.com/liuzuxin/OSRL

[2]https://github.com/young-geng/CQL

[3]https://github.com/guanjiayi/voce

[4]https://github.com/ZhengYinan-AIR/FISOR

[5]https://github.com/qianlin04/Safe-offline-RL-with-diffusion-model

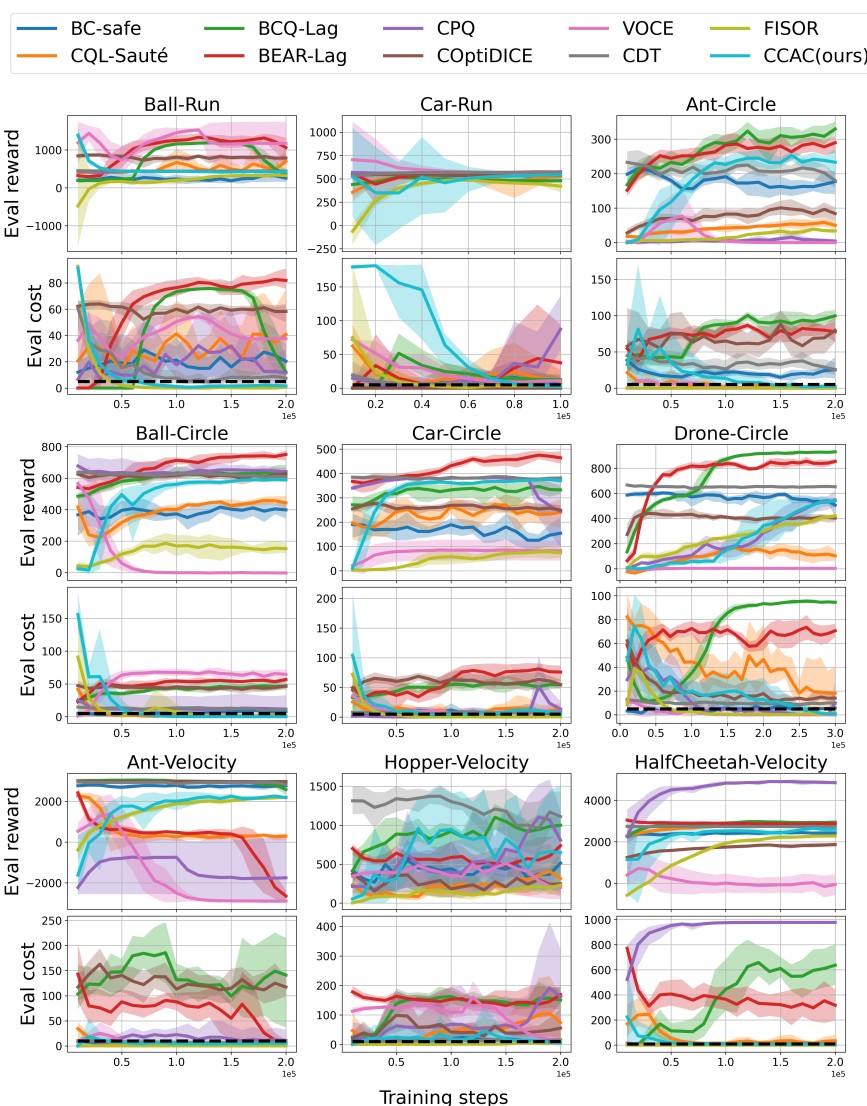

Figure 9: Training curves. The dashed line represents the cost threshold, which is 5 for `Run` and `Circle` tasks and 10 for `Velocity` tasks. The solid line and the light shade area represent the mean and mean±standard deviation. The interaction data is not added to the offline dataset.

COptiDICE often attain high rewards across various tasks, but this is accompanied by elevated costs. While FISOR ensures safety, it yields low rewards, achieving only about half of what our method can achieve. In contrast, our method can balance the rewards and costs as we analyzed in sections 5.

**Sample efficiency.** We can also observe that in certain environments, such as Car-Run and Drone-Circle, CCAC takes more steps to converge. However, it is important to note that we are working in the offline safe RL setting where training occurs solely on a fixed offline dataset Taking more steps to converge does not imply additional safety violations as there is no interaction with the environment during training. Besides, evaluation during training is optional, and Figure 9 is provided solely as a reference to illustrate the training progress. Furthermore, while sample efficiency is valuable, a method that achieves efficiency at the expense of safety may be less meaningful. Figure 9 shows that although some baselines converge faster, such as FISOR in Car-Run and BCQ-Lag in Drone-Circle, they either fail to achieve high rewards or cannot maintain safety after convergence. In contrast, our method consistently achieves both high reward and safety after convergence.

## D.2 COMPLETE RESULTS FOR DIFFERENT DATA PERCENTAGES

The detailed evaluation results for different dataset percentages are presented in Table 4. For baselines, we focus on BC-safe, CQL-Sauté, and FISOR. BC-safe helps gauge the difficulty of a task. If mimicking the safe actions in the datasets achieves both safety and high rewards, then the task is relatively easy. CQL-Sauté is adopted from (Sootla et al., 2022) by combining the conservatism estimation technique of (Kumar et al., 2020) into the offline setting. This baseline also employs the cost-to-go, like our method, but focuses solely on maximizing rewards. FISOR is the only baseline method evaluated that ensures safety. The results show that our method and FISOR consistently stay within the cost threshold across all tasks, with our method displaying higher rewards in all tasks. We can also observe that in most tasks, rewards decrease, costs increase, and variance rises as the percentage of data being used to train the methods decreases, as less training data typically leads to greater distribution shifts during testing. FISOR achieves this through its conservative approach while we achieve this by data augmentation and OOD detection.

| Methods | Ball-Run | | Car-Run | | Ant-Circle | | Ball-Circle | | Car-Circle | | Drone-Circle | | Average | |
|---|---|---|---|---|---|---|---|---|---|---|---|---|---|---|
| | Reward ↑ | Cost ↓ | Reward ↑ | Cost ↓ | Reward ↑ | Cost ↓ | Reward ↑ | Cost ↓ | Reward ↑ | Cost ↓ | Reward ↑ | Cost ↓ | Reward ↑ | Cost ↓ |
| Ours ($p=1.0$) | **0.97±0.01** | **0.27±0.19** | **0.95±0.04** | **0.19±0.27** | **1.01±0.26** | **0.55±1.57** | **0.87±0.03** | **0.0±0.0** | **0.85±0.04** | **0.73±1.95** | **0.82±0.11** | **0.07±0.54** | **0.91±0.14** | **0.3±1.09** |
| Ours ($p=0.75$) | **0.97±0.01** | **0.27±0.19** | **0.95±0.02** | **0.13±0.1** | **1.0±0.33** | **0.66±1.62** | **0.87±0.05** | **0.0±0.0** | **0.84±0.05** | **0.85±2.7** | **0.79±0.18** | **0.0±0.0** | **0.9±0.17** | **0.32±1.33** |
| Ours ($p=0.5$) | **0.96±0.07** | **0.33±0.19** | **0.97±0.01** | **0.3±0.77** | **0.9±0.38** | **0.83±1.75** | **0.88±0.02** | **0.0±0.0** | **0.85±0.05** | **0.96±3.0** | **0.76±0.19** | **0.68±2.09** | **0.89±0.19** | **0.52±1.72** |
| Ours ($p=0.25$) | **0.94±0.03** | **0.39±0.25** | **0.93±0.01** | **0.36±0.23** | **0.83±0.37** | **0.93±2.51** | **0.82±0.04** | **0.22±0.34** | **0.87±0.04** | 1.1±2.35 | **0.72±0.19** | **0.96±4.33** | **0.84±0.21** | **0.66±2.22** |
| FISOR ($p=1.0$) | **0.67±0.19** | **0.0±0.0** | **0.78±0.12** | **0.0±0.0** | **0.15±0.14** | **0.0±0.0** | **0.25±0.11** | **0.0±0.0** | **0.25±0.17** | **0.0±0.0** | **0.63±0.10** | **0.0±0.0** | **0.45±0.28** | **0.0±0.0** |
| FISOR ($p=0.75$) | **0.68±0.23** | **0.0±0.0** | **0.83±0.08** | **0.0±0.0** | **0.16±0.14** | **0.0±0.0** | **0.37±0.11** | **0.0±0.0** | **0.21±0.21** | **0.0±0.0** | **0.68±0.21** | **0.0±0.0** | **0.49±0.31** | **0.0±0.0** |
| FISOR ($p=0.5$) | **0.58±0.17** | **0.0±0.0** | **0.72±0.19** | **0.0±0.0** | **0.16±0.12** | **0.0±0.0** | **0.39±0.14** | **0.0±0.0** | **0.31±0.22** | **0.0±0.0** | **0.69±0.11** | **0.0±0.0** | **0.47±0.26** | **0.0±0.0** |
| FISOR ($p=0.25$) | **0.58±0.12** | **0.0±0.0** | **0.74±0.12** | **0.02±0.03** | **0.16±0.13** | **0.0±0.0** | **0.41±0.11** | **0.0±0.0** | **0.25±0.20** | **0.0±0.0** | **0.74±0.12** | **0.0±0.0** | **0.48±0.26** | **0.0±0.0** |
| BC-safe ($p=1.0$) | 0.36±0.32 | 2.08±3.63 | 0.96±0.03 | 1.06±2.04 | 0.56±0.33 | 3.5±5.73 | 0.62±0.19 | 1.67±1.52 | **0.38±0.27** | **0.58±1.46** | 0.82±0.16 | 1.32±1.49 | 0.62±0.33 | 1.7±3.21 |
| BC-safe ($p=0.75$) | 0.46±0.39 | 3.66±4.8 | **0.94±0.07** | **0.11±0.43** | 0.57±0.27 | 4.26±4.5 | 0.59±0.17 | 1.68±1.73 | 0.43±0.3 | 1.53±3.14 | 0.78±0.24 | 1.46±1.91 | 0.63±0.32 | 2.12±3.46 |
| BC-safe ($p=0.5$) | 0.6±0.69 | 3.63±5.13 | **0.84±0.15** | **0.44±0.69** | 0.57±0.26 | 4.66±5.19 | 0.59±0.2 | 2.32±1.89 | 0.42±0.27 | 1.09±4.9 | 0.77±0.22 | 2.01±1.75 | 0.63±0.38 | 2.36±4.01 |
| BC-safe ($p=0.25$) | 0.61±0.82 | 5.09±7.17 | **0.78±0.2** | **0.31±0.54** | 0.5±0.29 | 4.37±5.47 | 0.54±0.22 | 1.73±1.86 | 0.32±0.29 | 1.29±6.04 | 0.67±0.22 | 3.29±3.07 | 0.57±0.43 | 2.68±4.97 |
| CQL-Sauté ($p=1.0$) | 1.95±0.94 | 9.26±6.33 | 0.93±0.04 | 0.67±1.24 | **0.2±0.11** | **0.0±0.0** | **0.65±0.13** | **0.02±0.18** | 0.54±0.26 | 2.84±3.8 | 0.15±0.1 | 2.49±7.06 | 0.74±0.73 | 2.55±5.28 |
| CQL-Sauté ($p=0.75$) | 1.48±0.8 | 6.56±5.34 | **0.97±0.01** | **0.85±1.38** | **0.21±0.13** | **0.0±0.0** | **0.69±0.06** | **0.04±0.14** | 0.51±0.19 | 1.92±4.66 | 0.21±0.1 | 4.02±5.75 | 0.68±0.56 | 2.23±4.45 |
| CQL-Sauté ($p=0.5$) | 1.63±0.56 | 8.47±4.2 | 0.95±0.04 | 1.84±2.12 | **0.2±0.14** | **0.32±2.36** | **0.67±0.05** | **0.02±0.09** | 0.43±0.19 | 2.1±5.34 | 0.17±0.11 | 2.82±6.54 | 0.68±0.56 | 2.6±4.94 |
| CQL-Sauté ($p=0.25$) | 1.97±1.09 | 12.44±4.95 | **0.95±0.02** | **0.1±0.11** | **0.21±0.12** | **0.8±3.68** | 0.67±0.08 | 1.11±2.48 | **0.52±0.18** | **0.87±3.11** | 0.16±0.08 | 3.21±7.21 | 0.75±0.76 | 3.09±6.0 |

Table 4: Evaluation results of the normalized reward and cost under different data percentages. The cost threshold is 1. $p = 1.0/0.75/0.5/0.25$ means 100%, 75%, 50%, and 25% of the offline data is used during training respectively. ↑: the higher the reward, the better. ↓: the lower the cost (up to the threshold 1), the better. Each value is averaged over 20 episodes and 3 seeds. **Bold**: Safe agents whose normalized cost is smaller than 1. Gray: Unsafe agents.

## D.3 ADAPTATION RESULTS OF UNSEEN COST BUDGETS

We train the agents using the dataset shown in Figure 8 with safe trajectories being removed. During evaluation, the cost budget is set to be $[3, 4, 5, 6, 7, 8, 9]$ for Run and Circle tasks and $[3, 6, 9, 12, 15, 18]$ for Velocity tasks. The adaptation results are shown in Figure 10. We can observe that our method is capable of adapting to different unseen cost budgets while the baselines either struggle to maintain safety or display overly conservative behaviors, resulting in low rewards.

Table 5 shows the detailed results of Table 2. The agents are also trained using the dataset shown in Figure 8. For evaluation, the cost budget is 5 for Run and Circle tasks and 10 for Velocity tasks. It is important to note that for CQL-Sauté, CDT and our method, during training, they have no access to the cost budget that will be used to evaluate them. In contrast, for the rest of the baselines, although they are trained using the same cost budget as evaluation, the safe trajectories are removed, requiring them to display some sort of adaptability to learn safe policies from unsafe datasets. Compared with the results in Table 1, where the agents are trained using the whole dataset, we can observe that the number of safe agents decreases, and most of the methods achieve higher costs since the remaining dataset consists of unsafe trajectories with higher costs. However, our method still achieves the best performance among all the methods in terms of safety and high rewards.

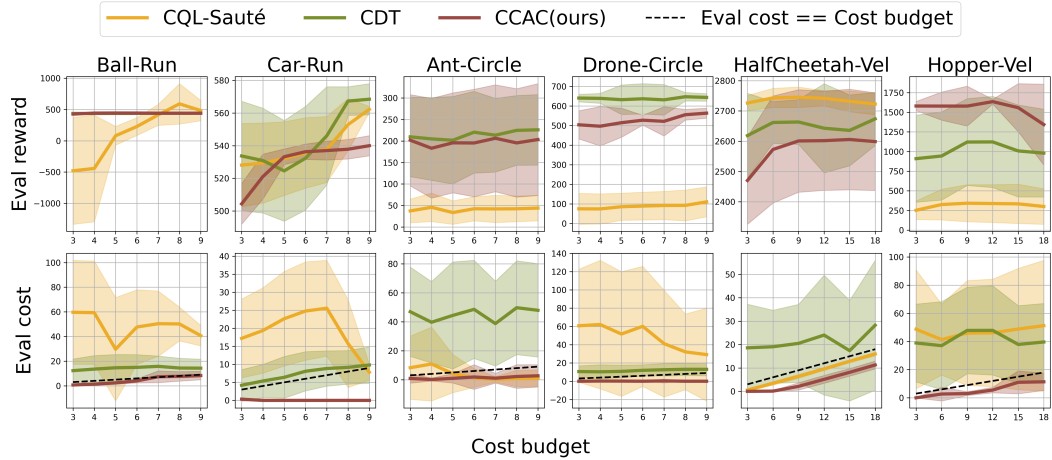

Figure 10: Results of adaptation to different unseen cost budgets. The agents are trained on a dataset where safe trajectories with cumulative costs smaller than a certain threshold are removed. The top-row plots show the evaluated reward and the bottom-row plots show the evaluated cost. The solid line and the light shade area represent the mean and mean±standard deviation. The dashed line represents the scenario where the evaluated cost matches the cost budget.

| Taks | Metric | CQL-Saute | BCQ-Lag | BEAR-Lag | CPQ | COptiDICE | VOCE | CDT | FISOR | CCAC(ours) |
|---|---|---|---|---|---|---|---|---|---|---|
| Ball-Run | Reward ↑ | $0.18_{\pm0.33}$ | $1.98_{\pm1.1}$ | $2.88_{\pm0.18}$ | $0.96_{\pm0.03}$ | $1.87_{\pm0.13}$ | $2.37_{\pm1.13}$ | $0.98_{\pm0.02}$ | $\mathbf{0.74_{\pm0.03}}$ | $\mathbf{0.99_{\pm0.02}}$ |
| | Cost ↓ | $5.93_{\pm8.39}$ | $10.36_{\pm7.27}$ | $16.82_{\pm1.28}$ | $2.73_{\pm3.19}$ | $12.55_{\pm0.61}$ | $12.55_{\pm8.84}$ | $1.59_{\pm0.81}$ | $\mathbf{0.34_{\pm1.30}}$ | $\mathbf{0.47_{\pm0.19}}$ |
| Car-Run | Reward ↑ | $0.93_{\pm0.04}$ | $1.04_{\pm0.02}$ | $0.89_{\pm0.33}$ | $0.95_{\pm0.02}$ | $\mathbf{0.89_{\pm0.02}}$ | $0.93_{\pm0.02}$ | $0.99_{\pm0.02}$ | $\mathbf{0.74_{\pm0.10}}$ | $\mathbf{0.93_{\pm0.01}}$ |
| | Cost ↓ | $4.53_{\pm2.62}$ | $29.34_{\pm9.62}$ | $30.29_{\pm5.93}$ | $1.12_{\pm1.64}$ | $\mathbf{0.0_{\pm0.0}}$ | $6.15_{\pm7.85}$ | $1.09_{\pm0.56}$ | $\mathbf{0.74_{\pm2.33}}$ | $\mathbf{0.0_{\pm0.0}}$ |
| Ant-Circle | Reward ↑ | $\mathbf{0.13_{\pm0.11}}$ | $1.27_{\pm0.29}$ | $1.14_{\pm0.34}$ | $\mathbf{0.01_{\pm0.03}}$ | $0.36_{\pm0.27}$ | $\mathbf{0.0_{\pm0.0}}$ | $0.87_{\pm0.32}$ | $\mathbf{0.09_{\pm0.08}}$ | $\mathbf{0.76_{\pm0.45}}$ |
| | Cost ↓ | $\mathbf{0.84_{\pm2.63}}$ | $19.98_{\pm6.82}$ | $17.03_{\pm8.59}$ | $\mathbf{0.0_{\pm0.0}}$ | $15.17_{\pm17.76}$ | $\mathbf{0.0_{\pm0.0}}$ | $5.32_{\pm3.58}$ | $\mathbf{0.0_{\pm0.0}}$ | $\mathbf{0.19_{\pm0.64}}$ |
| Ball-Circle | Reward ↑ | $0.68_{\pm0.13}$ | $0.99_{\pm0.13}$ | $1.13_{\pm0.11}$ | $1.0_{\pm0.11}$ | $0.94_{\pm0.06}$ | $0.0_{\pm0.01}$ | $0.96_{\pm0.04}$ | $\mathbf{0.31_{\pm0.13}}$ | $\mathbf{0.86_{\pm0.03}}$ |
| | Cost ↓ | $5.97_{\pm3.97}$ | $11.42_{\pm2.81}$ | $11.24_{\pm1.34}$ | $5.0_{\pm5.35}$ | $10.2_{\pm1.64}$ | $32.56_{\pm12.79}$ | $2.0_{\pm0.5}$ | $\mathbf{0.0_{\pm0.0}}$ | $\mathbf{0.0_{\pm0.0}}$ |
| Car-Circle | Reward ↑ | $0.68_{\pm0.31}$ | $0.81_{\pm0.28}$ | $1.0_{\pm0.1}$ | $0.96_{\pm0.07}$ | $0.62_{\pm0.05}$ | $0.26_{\pm0.04}$ | $0.85_{\pm0.08}$ | $\mathbf{0.26_{\pm0.18}}$ | $\mathbf{0.8_{\pm0.04}}$ |
| | Cost ↓ | $10.36_{\pm8.03}$ | $11.68_{\pm5.17}$ | $10.11_{\pm6.66}$ | $9.21_{\pm8.1}$ | $11.83_{\pm5.71}$ | $7.65_{\pm9.3}$ | $1.54_{\pm0.97}$ | $\mathbf{0.0_{\pm0.0}}$ | $\mathbf{0.45_{\pm1.39}}$ |
| Drone-Circle | Reward ↑ | $0.13_{\pm0.1}$ | $1.37_{\pm0.02}$ | $1.24_{\pm0.05}$ | $\mathbf{0.59_{\pm0.24}}$ | $0.58_{\pm0.1}$ | $\mathbf{0.0_{\pm0.0}}$ | $0.95_{\pm0.03}$ | $\mathbf{0.44_{\pm0.12}}$ | $\mathbf{0.76_{\pm0.11}}$ |
| | Cost ↓ | $10.34_{\pm13.65}$ | $19.38_{\pm1.21}$ | $14.39_{\pm1.78}$ | $\mathbf{0.73_{\pm4.08}}$ | $2.08_{\pm1.86}$ | $\mathbf{0.0_{\pm0.0}}$ | $1.45_{\pm0.79}$ | $\mathbf{0.70_{\pm1.39}}$ | $\mathbf{0.03_{\pm0.26}}$ |
| Ant-Velocity | Reward ↑ | $\mathbf{0.1_{\pm0.16}}$ | $0.7_{\pm0.34}$ | $-0.72_{\pm0.48}$ | $\mathbf{-0.96_{\pm0.14}}$ | $1.05_{\pm0.01}$ | $\mathbf{-1.02_{\pm0.0}}$ | $0.99_{\pm0.0}$ | $\mathbf{0.47_{\pm0.12}}$ | $\mathbf{0.71_{\pm0.24}}$ |
| | Cost ↓ | $\mathbf{0.41_{\pm0.52}}$ | $17.82_{\pm9.51}$ | $\mathbf{0.0_{\pm0.0}}$ | $\mathbf{0.0_{\pm0.0}}$ | $10.83_{\pm5.37}$ | $\mathbf{0.0_{\pm0.0}}$ | $1.14_{\pm0.59}$ | $\mathbf{0.0_{\pm0.0}}$ | $\mathbf{0.39_{\pm0.2}}$ |
| HalfCheetah-Velocity | Reward ↑ | $\mathbf{1.01_{\pm0.01}}$ | $1.09_{\pm0.07}$ | $1.08_{\pm0.01}$ | $1.83_{\pm0.08}$ | $\mathbf{0.69_{\pm0.02}}$ | $-0.2_{\pm0.01}$ | $0.98_{\pm0.03}$ | $\mathbf{0.77_{\pm0.02}}$ | $\mathbf{0.96_{\pm0.06}}$ |
| | Cost ↓ | $\mathbf{0.71_{\pm0.09}}$ | $52.22_{\pm39.86}$ | $54.14_{\pm13.19}$ | $97.88_{\pm0.19}$ | $\mathbf{0.0_{\pm0.0}}$ | $\mathbf{0.0_{\pm0.0}}$ | $2.28_{\pm1.85}$ | $\mathbf{0.0_{\pm0.0}}$ | $\mathbf{0.3_{\pm0.16}}$ |
| Hopper-Velocity | Reward ↑ | $0.2_{\pm0.14}$ | $0.54_{\pm0.35}$ | $0.17_{\pm0.22}$ | $0.35_{\pm0.16}$ | $0.21_{\pm0.15}$ | $\mathbf{0.01_{\pm0.01}}$ | $0.65_{\pm0.32}$ | $\mathbf{0.15_{\pm0.05}}$ | $\mathbf{0.91_{\pm0.14}}$ |
| | Cost ↓ | $5.11_{\pm4.14}$ | $11.73_{\pm4.62}$ | $6.16_{\pm7.75}$ | $7.66_{\pm7.56}$ | $9.98_{\pm7.41}$ | $\mathbf{0.0_{\pm0.0}}$ | $5.32_{\pm3.41}$ | $\mathbf{0.25_{\pm0.43}}$ | $\mathbf{0.44_{\pm0.19}}$ |
| Average | Reward ↑ | $0.45_{\pm0.4}$ | $1.09_{\pm0.58}$ | $0.98_{\pm0.93}$ | $0.63_{\pm0.75}$ | $0.8_{\pm0.47}$ | $0.26_{\pm0.96}$ | $0.91_{\pm0.19}$ | $\mathbf{0.44_{\pm0.27}}$ | $\mathbf{0.85_{\pm0.21}}$ |
| | Cost ↓ | $4.91_{\pm7.33}$ | $20.44_{\pm19.33}$ | $17.8_{\pm16.46}$ | $13.81_{\pm30.22}$ | $8.07_{\pm8.85}$ | $6.54_{\pm12.1}$ | $2.41_{\pm2.45}$ | $\mathbf{0.22_{\pm1.05}}$ | $\mathbf{0.25_{\pm0.56}}$ |

Table 5: Complete evaluation results of the normalized reward and cost of agents trained on datasets with partial data removed. The cost threshold is 1 (the cost budget is 5 for `Run` and `Circle` tasks and 10 for `Velocity` task). ↑: the higher the reward, the better. ↓: the lower the cost (up to 1), the better. Each value is averaged over 20 episodes and 3 seeds. **Bold**: Safe agents. Gray: Unsafe agents. **Blue**: Safe agent with the highest reward.

## D.4 EVALUATING POLICY REGULARIZATION VIA IQL INTEGRATION

To evaluate whether our method achieves a similar policy regularization effect, we construct the following experiment. We adopt an existing method, IQL (Kostrikov et al., 2021), which learns $Q_r(s,a)$ and $V(s)$ by using expectile regression with a SARSA-style objective to avoid sampling OOD actions; thus, the learned $Q_r(s,a)$ represents the value of the behavior policy. Then we replace Eq.(7) with IQL for learning $Q_r(s,a|\kappa)$, while keeping all other components unchanged. The results in Table 6 demonstrate that integrating IQL has little impact on performance, which provides further

support that the overestimation of $Q_c(s, a|\kappa)$ discourages OOD actions in a manner similar to policy regularization (without the direct regularization of $Q_r(s, a|\kappa)$).

| Task | Metric | CCAC(in Table 1) | CCAC(IQL) |
|------|--------|------------------|-----------|
| Ball-Run | Reward ↑ | $0.97 \pm 0.01$ | $0.98 \pm 0.03$ |
| | Cost ↓ | $0.27 \pm 0.19$ | $0.47 \pm 0.19$ |
| Ball-Circle | Reward ↑ | $0.87 \pm 0.03$ | $0.84 \pm 0.02$ |
| | Cost ↓ | $0.0 \pm 0.0$ | $0.0 \pm 0.0$ |
| Car-Circle | Reward ↑ | $0.85 \pm 0.04$ | $0.8 \pm 0.05$ |
| | Cost ↓ | $0.73 \pm 1.95$ | $0.96 \pm 2.29$ |
| Ant-Velocity | Reward ↑ | $0.9 \pm 0.05$ | $0.87 \pm 0.02$ |
| | Cost ↓ | $0.58 \pm 0.15$ | $0.44 \pm 0.11$ |
| HalfCheetah-Velocity | Reward ↑ | $0.96 \pm 0.04$ | $0.99 \pm 0.02$ |
| | Cost ↓ | $0.79 \pm 0.2$ | $0.3 \pm 0.23$ |
| Average | Reward ↑ | $0.91 \pm 0.07$ | $0.9 \pm 0.08$ |
| | Cost ↓ | $0.47 \pm 0.93$ | $0.43 \pm 1.07$ |

Table 6: Evaluation results of the normalized reward and cost. ↑: the higher the reward, the better. ↓: the lower the cost (up to 1), the better. Each value is averaged over 20 episodes and 3 seeds.

