# OpenReview forum: "Constraint-Conditioned Actor-Critic for Offline Safe Reinforcement Learning"
_ICLR.cc/2025/Conference — ICLR 2025 Poster_

### Official Review · Reviewer_GJjh · 2024-10-17

**Soundness:** 3
**Presentation:** 3
**Contribution:** 3
**Rating:** 8
**Confidence:** 3

**Summary:**

This paper addresses the challenge of safe offline reinforcement learning (RL) and introduces a novel approach termed Constraint Conditioned Actor Critic (CCAC). CCAC is designed to facilitate adaptive policy learning under various constraint limits while ensuring high rewards and minimizing distributional shifts. To achieve this, CCAC employs a cost-to-goal conditional Variational Autoencoder (CVAE) to model the adaptive behavior across different safety constraint intensities. Additionally, a discriminator is utilized to autonomously evaluate the safety levels. By integrating the discriminator with the CVAE, CCAC establishes an Out-of-Distribution (OOD) detector, \( v(s,a|k) \), that assesses both distributional shift and safety performance concurrently. Utilizing this framework, CCAC executes standard actor-critic updates to develop a constraint-aware critic and policy. The effectiveness of CCAC has been demonstrated through evaluations on the DSRL benchmark and extensive ablation studies.

**Strengths:**

1. The concept of CCAC is intuitively straightforward and easy to implement.

2. This paper is well-written, clearly highlighting the challenges, motivations and contributions.

3. CCAC achieves SOTA performance in several DSRL envs.

4. The ablation studies are comprehensive, clearly demonstrating the advantages of CCAC over other baseline methods. Notably, CCAC exhibits constraint-aware behavior that guarantees strong performance (high rewards and low costs below the constraint limit). Furthermore, CCAC demonstrates robustness across varying quantities of data.

**Weaknesses:**

Several minor weaknesses are present in the study. Addressing these concerns could significantly improve my evaluation of the manuscript:

1. Limited evaluation environments. The DSRL benchmark offers over 30 environments, but this paper only conducts experiments on solely 9 envs. Including more scenarios can substantially enhance the paper quality.

2. Some assumption. CCAC utilizes the trained CVAE to augment the dataset and annotates these data with their original cost/reward values. While this approach is acceptable, it may introduce some errors if the reward/cost functions are not smooth across the spatial space, which then will introduce some annotating errors.

3. The overconservative behavior of FISOR is primarily attributed to the use of hard constraint, rather than its use of the largest feasible region as claimed by the authors in line 323. In fact, using the largest feasible region will include more actionable actions for policy optimization and so can reduce the overconservatism.

**Questions:**

See weaknesses for details.

---

> ### Author Response · Authors · 2024-11-16
> **Rebuttal by Authors**
>
> We thank the reviewer for your thoughtful comments and constructive feedback. We address each weakness and question as follows.
>
> ### Weaknesses and Questions
> > Limited evaluation environments. The DSRL benchmark offers over 30 environments, but this paper only conducts experiments on solely 9 envs. Including more scenarios can substantially enhance the paper quality.
>
> We agree with the reviewer that additional experiments would be beneficial. However, we would like to emphasize that our method is formulated within the Constrained MDP framework, which assumes access to the full state. To maintain consistency, we consider environments that satisfy this requirement, such as Run, Circle, and Velocity. In contrast, the other environments (Goal, Button, Push, and Metadrive) provide only partial observations, such as pseudo-lidar data, rather than full state information, making them less aligned with the CMDP framework and therefore outside the scope of this paper. Although some baselines (e.g., BCQ-Lag, BEAR-Lag, CPQ, COptiDICE, CDT, FISOR) have been directly evaluated in Goal, Button, Push, and Metadrive by treating the observations as states [1, 2], it can be observed that they produce suboptimal behaviors, either being overly conservative or struggling to maintain safety. In our opinion, these baselines and our method are not designed for partially observable settings, and such comparisons would not provide meaningful insights. \
> [1] https://arxiv.org/pdf/2306.09303 \
> [2] https://arxiv.org/pdf/2401.10700
>
> > Some assumption. CCAC utilizes the trained CVAE to augment the dataset and annotates these data with their original cost/reward values. While this approach is acceptable, it may introduce some errors if the reward/cost functions are not smooth across the spatial space, which then will introduce some annotating errors.
>
> We acknowledge that the trained CVAE may introduce some errors, as it cannot perfectly model the constraint-conditioned state-action distribution. However, these errors are minor and have a limited impact on our method's overall performance. As explained in lines 215-219, overestimating $Q_c$ values for OOD state-action pairs can result in a mildly conservative policy that naturally avoids risky actions. Additionally, it is important to note that the cost functions in our training and evaluation environments are not smooth, with a cost of 1 when a constraint is violated (e.g., when an agent's position exceeds a safety boundary) and 0 otherwise. The results in Sections 5.1 and 5.2 show that our method consistently achieves both safety and high rewards, indicating that the errors introduced by the CVAE are effectively mitigated. Furthermore, our ablation study in Section 5.3 underscores the importance of the CVAE, as removing it leads to significant performance degradation.
>
> > The overconservative behavior of FISOR is primarily attributed to the use of hard constraint, rather than its use of the largest feasible region as claimed by the authors in line 323. In fact, using the largest feasible region will include more actionable actions for policy optimization and so can reduce the overconservatism.
>
> Thank you for highlighting this point. We have revised the statement accordingly to clarify this aspect.

---

> ### Author Response · Authors · 2024-11-21
> **Reminder for Re-evaluation and Further Discussion**
>
> Dear reviewer,
>
> Thanks again for your valuable comments. We hope our new response has addressed your concerns, and we would greatly appreciate it if you could re-evaluate our submission based on the response, or let us know whether you have any other questions. Thanks!
>
> Best, \
> Authors

---

> > ### Comment · Reviewer_GJjh · 2024-11-29
> > **Thanks for the detailed responses**
> >
> > I appreciate the authors detailed rebuttal. To me, offline safe RL seeks to strike a good balance across three objectives: maximizing reward, staying close to behavior policy, minimizing the cost, and this paper can find a good intersetion across them. Intuitively, CCAC filters out all out-of-distribution data and high cost data through the classifier and then seeks to maximize the reward based on these filtered data. So, I think this paper presents a good addition to the safe offline RL commutiny. Regarding the extreme case mentioned by the reviewer pD1A that "low cost and low value data dominates the dataset", I think in this case, most offline safe RL cannot obtain a good policy, especially for those relying on in-sample learning. Thus, I decided to raise my score.
> >
> > However, it would be good for the authors to discuss more differences about OOD and OOB, as mentioned by the reviewer pD1A. Also, I recently noticed a relevant work that should be carefully discussed in this paper [1], the authors should thoroughly discuss the techinical contribution compared to [1].
> >
> > [1] Adaptive Advantage-guided Policy Regularization for Offline Reinforcement Learning, ICML 2024.

---

> > > ### Author Response · Authors · 2024-12-01
> > >
> > > Thank you for your thoughtful feedback and for raising your score. We very much appreciate it.
> > >
> > > Regarding the differences between OOD and OOB, we kindly refer you to the discussion thread in response to reviewer pD1A.
> > >
> > > Regarding the "Adaptive Advantage-guided Policy Regularization" paper, we have already discussed it, albeit briefly, in response to reviewer REk7's comment in the second paragraph of the Related Work section (lines 100-102). We are happy to expand the discussion in future versions of this paper (the deadline to upload a revised manuscript during the response period has passed unfortunately). Basically the key differences are 1) our CVAE is conditioned on cost thersholds that allows us to generate not only actions but also states for augmentation when learning critics and policy; and 2) we learn a constriant conditioned critics, $Q_r(s, a | \kappa)$ and $Q_c(s, a | \kappa)$, and a policy, $\pi(s, a | \kappa)$, that offer the flexibility to adapt to varying thresholds rather than $Q(s, a)$ and $\pi(a | s)$ that lack such adaptability.

---

### Official Review · Reviewer_pD1A · 2024-10-18

**Soundness:** 2
**Presentation:** 3
**Contribution:** 2
**Rating:** 6
**Confidence:** 4

**Summary:**

This paper focused on offline safe RL. The authors proposed CCAC that can learn with adaptive constraint budgets. CCAC first trained a binary classifier to distinguish OOD state-action pairs with constraint budgets acting as labels. The OOD distribution was then used for constrained optimization for the actor and critic. The authors provided theoretical justification and comprehensive empirical investigation.

**Strengths:**

The paper is well-written in general. The idea is interesting and technical details easy to follow. The authors conducted extensive experiments showing CCAC performed favorably against the existing methds.

**Weaknesses:**

My concern is that the paper seems to blur the boundary between out-of-distribution and out-of-budget (OOB) state-action pairs. If we assume all OOD state-action pairs are unsafe, then it means the algorithm cannot generalize to unseen state-action pairs since their costs are lower bounded by $\epsilon$. This seems at odds with the standard offline RL setting which seeks to generalize and improve upon the behavior policy and beyond the dataset.
If we recall that the key issue of offline RL lies in that incorrect action values of unseen state-action pairs cannot be corrected, then in my opinion mixing OOD and OOB runs the risk of not generalizing beyond the dataset $\mathcal{D}$ since safe unseen $(s,a)$ always have $Q_c \geq \epsilon$. If this is true, then later analysis like Theorem 4.1 is not very meaningful (which itself is an variant of CQL derivation imo).

**Questions:**

Please refer to the above for questions.

---

> ### Author Response · Authors · 2024-11-16
> **Rebuttal by Authors**
>
> We are grateful for the thoughtful comments provided by the reviewer. In the following, we address the weaknesses and questions in detail.
>
> ### Weaknesses and Questions
> 1. My concern is that the paper seems to blur the boundary between out-of-distribution and out-of-budget (OOB) state-action pairs. If we assume all OOD state-action pairs are unsafe, then it means the algorithm cannot generalize to unseen state-action pairs since their costs are lower bounded by $\epsilon$. This seems at odds with the standard offline RL setting which seeks to generalize and improve upon the behavior policy and beyond the dataset. If we recall that the key issue of offline RL lies in that incorrect action values of unseen state-action pairs cannot be corrected, then in my opinion mixing OOD and OOB runs the risk of not generalizing beyond the dataset $\mathcal{D$ since safe unseen $(s, a)$ always have $Q_c > \epsilon$. If this is true, then later analysis like Theorem 4.1 is not very meaningful (which itself is an variant of CQL derivation imo).
>
> We appreciate the reviewer's feedback. However, we also would like to clarify that this concern does not apply to our method. \
> **Definition of OOD in Our Method:** In our approach, we define OOD state-action pairs as those that violate constraints, as noted in lines 356-357 and 163-166. This is distinct from traditional offline RL, where OOD typically refers to unseen state-action pairs beyond the dataset. In our context, an unseen state-action pair is not necessarily OOD; it only becomes OOD if it violates the specified constraints. \
> **Learning Constraint-Conditioned Distributions:** In our method, we aim to learn the distribution of state-action pairs conditioned on constraints rather than action distribution given seen states in datasets like traditional offline RL. To achieve this, we introduce a CVAE to model this constriant-conditioned state-action distribution. While the CVAE can generate state-action pairs, it does not directly determine if the sampled pairs are OOD (i.e., violating constraints) or not. Thus, we propose a constraint-conditioned classifier to classify the generated samples. The CVAE and classifier together can approximate the constriant-conditioned distribution of OOD pairs as we discussed in lines 202-204.  \
> **$Q_c$ overestimation:** the trained CVAE and classifier are then used to construct OOD (i.e., violating constraint) state-action pairs, whose $Q_c$ values are overestimated during critics learning. Based on our definition and formulation, safe unseen state-action pairs are not considered OOD in our method. Thus, their $Q_c$ will not be overestimated, indicating that their $Q_c$ values are not greater than $\epsilon$. \
> **Generalization Results:** In Section 5, we evaluate the generalization performance of our method by 1) removing up to 75$\%$ percent of the data (Figure 2) and 2) filtering out the safe trajectories in the dataset whose cumulative cost is smaller than a certain threshold and then using unseen cost budgets for evaluation (Table 2). The emperical results have demonstrated that our method can consistently maintain safety and high rewards, which indicate that our method is capable of handling distribution shifts of states and actions and generalizing beyond the offline dataset.

---

> ### Comment · Reviewer_pD1A · 2024-11-16
> **significant clarification on the methodology needed**
>
> Hi, thanks for the clarification.
>
> I think this paper mixes the concept of out of distribution samples and out of budget constraint violation. This mixture may have worked to the advantage of getting good performance but I think significant explanation on the methodology is needed. Let me summarize the point on the setting:
>
> > The central issue in offline RL is OOD data can lead to erroneously high action value and the agent ends up choosing a bad action. We are interested in learning an agent that can generalize to unseen state-action pairs so their values will not be overestimated. This paper considers the CMDP setting, so in addition to rewards there is also constraint.  It is possible to have state-actions that have high action value but also high risk. The goal of offline safe RL should be to learn an agent that can generalize to unseen data so their action value $Q_r(s,a)$ and constraint violation $Q_c(s,a)$ are both properly estimated. There may be good OOD data with high value low risk. By learning from the dataset, we hope the agent can make a sensible decision to utilize these OOD data in such cases.
>
> Now let me rephrase my question. This paper seems to mix the OOD and OOB data. An OOD state action pair can have high value low risk. They are not equivalent to OOB data which can be (high value, high risk) or (low value, high risk). Key to the proposal is to regard all OOB samples as OOD, and train a generative model $\nu(s,a|\kappa)$ and a classfier $h(s,a | \kappa)$ to assign OOD labels. As the authors put it, this definition is distinct from prior works, so I think the authors need to carefully elaborate on the logic instead of detailing only technical details such as CVAE or binary classifier.
>
> IMO, the problem of assigning OOB as OOD is that, the OOD label assigned by $h(s,a|\kappa)$ is determined only by whether or not the cumulative constraint exceeding the threshold. Even if the label is 0, it does not really mean the sample is not OOD,
>  but actually only OOB. So what is presented to the agent is a tuple (?, low risk), that the agent has no way to know whether the value is high or low. Therefore, it seems mixing these two concepts still run the risk of the original OOD issue.

---

> ### Author Response · Authors · 2024-11-17
> **Clarification on the Methodology**
>
> We thank the reviewer for the additional feedback. Regarding "Even if the label is 0, it does not really mean the sample is not OOD, but actually only OOB." Did the reviewer mean "actually only **not** OOB" instead of "actually only OOB"? In our work, if the label is 0, it means that it is not OOB.
>
> We would like to emphasize that our work addresses **safe** offline RL with varying cost thresholds and not just offline RL. To this end, we introduce a new notion of OOD centered around constraint thresholds (or OOB, using the reviewer's terminology), and propose to use a constraint-conditioned VAE and classifier to model this OOD state-action distribution and augment the dataset (as described in Section 4.1). As demonstrated in our experimental results (Figure 2 and Table 2), this treatment of OOB data allows us to produce safe policies even for unseen cost thresholds and when the percentage of safe trajectories in the dataset is very small. In addition, although this does not directly handle the type of OOD data described in the reviewer's comment, our method performs indirect policy regularization to mitigate the OOD issue. As described in Section 4.3, overestimating the $Q_c$-values for OOB data leads to a policy that avoids taking risky actions and visiting unsafe states (as OOB state-action pairs imply either unsafe states or risky actions). This overestimation encourages the learned policy to remain close to the behavior policies (the dataset distributions), as the trajectories in the offline dataset are labeled with their cumulative costs and thus by design satisfy those cost constraints. Also, because this overestimation is not applied to the IND data, we avoid learning an overly conservative policy.

---

> ### Author Response · Authors · 2024-11-21
> **Reminder for Re-evaluation and Further Discussion**
>
> Dear reviewer,
>
> Thank you again for your thoughtful feedback. We hope our response has addressed your questions and concerns. If you have any further questions, we’d be happy to address them. If our response has resolved your concerns, we kindly ask you to consider raising your score. Thank you!
>
> Best, \
> Authors

---

> > ### Comment · Reviewer_pD1A · 2024-11-25
> >
> > Hi,
> >
> > I am rather confused by the authors' repetitive emphasis that
> > > We would like to emphasize that our work addresses safe offline RL with varying cost thresholds and not just offline RL.
> >
> > Safe RL cares about maximizing reward while subject to safety constraint. Since this paper focuses on the general CMDP setting, there can be many state-action pairs with low risk but also low values. If the authors were to consider the special setting where reward encodes safety, then regarding all OOB actions as OOD may still be reasonable. As I have pointed out in the first comment, the current methodology could have worked to the advantage of getting good performance when it is the case, but I don't think it will continue to make sense for all cases especially when low-risk low-value actions dominate the trajectories.
> >
> > IMO, the offline safe RL considered in this paper is actually flawed in both of its components: in terms of offline, it runs the original risk of overestimating value of actions and therefore divergence; for safety, it could well output actions that are trivially safe but with low reward. And I don't think this issue is minor and can be left to future work.

---

> ### Author Response · Authors · 2024-11-26
> **Additional clarifications and results regarding the OOD issue: Part I**
>
> 1. OOD issue
>
> While our method does not directly deal with the (low value, low risk) data the same way that we deal with (?, high risk) data with a CVAE and a classifier, it mitigates the OOD issue through **policy regularization**. Policy regularization is common method to address the OOD issue in offline RL. It penalizes deviations of the learned policy from the behavior policy [1, 2, 3]. For instance, CQL[1] learns a conservative (underestimation of) $Q_r$ to regularize the learned policy to stay close to the behavior policy. It is proved in Dual RL [4] that adding such constraint on value functions is equivalent to adding a $f$-divergence between the learned policy and behavior policy. Leveraging the dual relationship between $Q_r$ and $Q_c$ [5, 6], our method achieve policy regularization by overestimating $Q_c$ to make the learned policy stay close to the behavior policy (as mentioned in our previous response). Thus, when learning the $Q_r$, our method does not need an additional explicit regularization. The "Distribution Shifts" section and Figure 2 on Page 7 of our paper already provides evidence that our method can effectively mitigate the OOD issue - reducing the percentage of training data (down to 25%) increases the number of unseen state-action pairs. We have also conducted two new experiments to further support this claim and the results are shown in the following parts.
>
> 2. Additional results
>
> To evaluate whether our method achieves the same policy regularization effect empirically, we construct the following experiment. We adopt an existing method, IQL [3], which learns $Q_r(s, a)$ and $V(s)$ by using expectile regression (Eq.(5) and (6) in [3]) with a SARSA-style objective to avoid sampling OOD actions; thus, the learned $Q_r(s, a)$ represents the value of the behavior policy. Then we replace Eq.(7) in our paper with Eq.(5) and (6) from IQL for learning $Q_r(s, a | \kappa)$, while keeping all other components unchanged. If "So what is presented to the agent is a tuple (?, low risk), that the agent has no way to know whether the value is high or low" were true, then integrating IQL should significantly improve performance (higher rewards), as IQL would lead to more accurate $Q_r(a | s, \kappa)$ estimates. However, the results demonstrate that integrating IQL has little impact on performance. This provides further support that the overestimation of $Q_c(s, a | \kappa)$ discourages OOD actions in a manner similar to policy regularization (without the direct regularization of $Q_r(s, a | \kappa)$).
>
> |         Task         |       Metric      | CCAC(in Table 1) |    CCAC(IQL)    |
> |:--------------------:|:-----------------:|:----------------:|:---------------:|
> |       Ball-Run       | Reward $\uparrow$ |  0.97 $\pm$ 0.01 | 0.98 $\pm$ 0.03 |
> |                      | Cost $\downarrow$ |  0.27 $\pm$ 0.19 | 0.47 $\pm$ 0.19 |
> |      Ball-Circle     | Reward $\uparrow$ |  0.87 $\pm$ 0.03 | 0.84 $\pm$ 0.02 |
> |                      | Cost $\downarrow$ |  0.0 $\pm$  0.0  |  0.0 $\pm$ 0.0  |
> |      Car-Circle      | Reward $\uparrow$ |  0.85 $\pm$ 0.04 |  0.8 $\pm$ 0.05 |
> |                      | Cost $\downarrow$ |  0.73 $\pm$ 1.95 | 0.96 $\pm$ 2.29 |
> |     Ant-Velocity     | Reward $\uparrow$ |  0.9 $\pm$ 0.05  | 0.87 $\pm$ 0.02 |
> |                      | Cost $\downarrow$ |  0.58 $\pm$ 0.15 | 0.44 $\pm$ 0.11 |
> | HalfCheetah-Velocity | Reward $\uparrow$ |  0.96 $\pm$ 0.04 | 0.99 $\pm$ 0.02 |
> |                      | Cost $\downarrow$ |  0.79 $\pm$ 0.2  |  0.3 $\pm$ 0.23 |
> |        Average       | Reward $\uparrow$ |  0.91 $\pm$ 0.07 |  0.9 $\pm$ 0.08 |
> |                      | Cost $\downarrow$ |  0.47 $\pm$ 0.93 | 0.43 $\pm$ 1.07 |
>
> [1] Kumar, Aviral, et al. "Conservative q-learning for offline reinforcement learning." Advances in Neural Information Processing Systems 33 (2020): 1179-1191. \
> [2] Lyu, Jiafei, et al. "Mildly conservative q-learning for offline reinforcement learning." Advances in Neural Information Processing Systems 35 (2022): 1711-1724. \
> [3] Kostrikov, Ilya, Ashvin Nair, and Sergey Levine. "Offline reinforcement learning with implicit q-learning." arXiv preprint arXiv:2110.06169 (2021). \
> [4] Sikchi, Harshit, et al. "Dual rl: Unification and new methods for reinforcement and imitation learning." arXiv preprint arXiv:2302.08560 (2023). \
> [5] Paternain, Santiago, et al. "Constrained reinforcement learning has zero duality gap." Advances in Neural Information Processing Systems 32 (2019). \
> [6] Paternain, Santiago, et al. "Safe policies for reinforcement learning via primal-dual methods." IEEE Transactions on Automatic Control 68.3 (2022): 1321-1336.

---

> ### Author Response · Authors · 2024-11-26
> **Additional clarifications and results regarding the OOD issue: Part II**
>
> 3. Dataset dominated by low-risk low-value trajectories
>
> The reviewer is actually describing an issue similar to class imbalance in classification. In the extreme, if the dataset contains *only* low-risk low-value trajectories, why should we expect any offline method to be able to learn a high-value policy? In any case, we have conducted the following additional experiment where we remove all the trajectories with rewards greater than half of the maximum reward among the trajectories with a cost less than some $\epsilon$ (basically removing the top half of the gray trajectories in Figure 8). Then we evaluate our method against a cost threshold of $0.5\epsilon$ (a more stringent constraint than $\epsilon$). Below are the results. We can observe that our method consistently learns safe policies with high (normalized) rewards (slighter lower than those in Table 1 but substantially higher than 0.5, basically performing in a regime where we have no such offline data). We point the reviewer to our explanation of this behavior ("stitching ability") at the top of page 9 of our paper.
>
> |         Task         |       Metric      | CCAC(in Table 1) |       CCAC      |
> |:--------------------:|:-----------------:|:----------------:|:---------------:|
> |       Ball-Run       | Reward $\uparrow$ |  0.97 $\pm$ 0.01 | 0.89 $\pm$ 0.02 |
> |                      | Cost $\downarrow$ |  0.27 $\pm$ 0.19 | 0.19 $\pm$ 0.17 |
> |        Car-Run       | Reward $\uparrow$ |  0.95 $\pm$ 0.04 |  0.9 $\pm$ 0.07 |
> |                      | Cost $\downarrow$ |  0.19 $\pm$ 0.27 | 0.07 $\pm$ 0.09 |
> |      Ant-Circle      | Reward $\uparrow$ |  1.01 $\pm$ 0.26 | 0.95 $\pm$ 0.23 |
> |                      | Cost $\downarrow$ |  0.55 $\pm$ 1.57 | 0.19 $\pm$ 0.92 |
> |      Ball-Circle     | Reward $\uparrow$ |  0.87 $\pm$ 0.03 | 0.87 $\pm$ 0.02 |
> |                      | Cost $\downarrow$ |  0.0 $\pm$  0.0  |  0.0 $\pm$ 0.0  |
> |      Car-Circle      | Reward $\uparrow$ |  0.85 $\pm$ 0.04 | 0.84 $\pm$ 0.03 |
> |                      | Cost $\downarrow$ |  0.73 $\pm$ 1.95 |  0.9 $\pm$ 2.18 |
> |     Drone-Circle     | Reward $\uparrow$ |  0.82 $\pm$ 0.11 | 0.77 $\pm$ 0.14 |
> |                      | Cost $\downarrow$ |  0.07 $\pm$ 0.54 |  0.0 $\pm$ 0.0  |
> |     Ant-Velocity     | Reward $\uparrow$ |  0.9 $\pm$ 0.05  |  0.8 $\pm$ 0.22 |
> |                      | Cost $\downarrow$ |  0.58 $\pm$ 0.15 | 0.67 $\pm$ 0.24 |
> | HalfCheetah-Velocity | Reward $\uparrow$ |  0.96 $\pm$ 0.04 | 0.97 $\pm$ 0.03 |
> |                      | Cost $\downarrow$ |  0.79 $\pm$ 0.2  |  0.54 $\pm$ 0.2 |
> |    Hopper-Velocity   | Reward $\uparrow$ |  0.89 $\pm$ 0.02 | 0.76 $\pm$ 0.22 |
> |                      | Cost $\downarrow$ |  0.32 $\pm$ 0.23 |  0.67 $\pm$ 1.0 |
> |        Average       | Reward $\uparrow$ |  0.91 $\pm$ 0.11 | 0.86 $\pm$ 0.16 |
> |                      | Cost $\downarrow$ |  0.39 $\pm$ 0.94 | 0.34 $\pm$ 0.83 |
>
> We would like to emphasize again that, in this paper, we focus on the challenging problem of safety in offline RL, that is, learning a poilcy that can adapt to different and potentially unseen and varying cost thresholds. This is a significant step beyond what we consider in a typical CMDP setting where the cost threshold is fixed. Our method learns adaptive policies through the novel use of constraint-conditioned VAE and classifier to handle "high-risk" data, as they are involved in constraint satisfaction, and implicit policy regularization via an overestimation of $Q_c$ (which also helps in constraint satisfaction) to achieve a high reward. Our method is the first value-based method that can achieve zero-shot adaptation to varying cost thresholds without the need of retraining and outperform existing state-of-the-art methods by a large margin.

---

> ### Comment · Reviewer_pD1A · 2024-11-29
>
> Thanks for the additional results, they are appreciated. I agree with reviewer GJjh that this paper offers something new. But I do not agree that my concerns are like an extreme case.
>
> To me what is the most critical point is really the positioning of the paper. As I said in the first comment,  I think the authors need to carefully elaborate on the logic. It is important to help the reader to understand what are the key quantities in play and how the authors deal with them. It is ok to have limitations, but it is important to acknowledge them to help better positioning. Because the authors opted for the CMDP setting, the constraint and reward need to be separately discussed. I cannot appreciate the original claim that seemed to advertise the unusual definition of OOB as OOD, because this mixing can cause much confusion. In the same spirit, the current phrasing also seems not straightforward and a bit overclaiming:
> > Although our definition of OOD differs from the typical definition of OOD which refers to unseen
> actions in the dataset, we can still handle the original OOD issue by overestimation of $Q_c(s, a|\kappa)$
> and the resulting effect of policy regularization.
>
> I would simply suggest the authors to clearly state the methodology and the downsides. This can likely be stated as "the method is positioned in the offline CMDP setting where the priority is to avoid dangerous OOB states, and if possible, achieve higher rewards. To this end, OOB states are regarded as OOD and new methods are introduced. While this method could potentially run the risk of the original OOD issues, the empirical results verify that well-balanced performance can be achieved..."
>
> I will consider raising my score if the authors can have a clear overarching paragraph that helps better position the paper and clearly explains the downsides.

---

> ### Author Response · Authors · 2024-12-01
>
> We thank the reviewer for the additional feedback and engaging us in this thoughtful discussion. We agree with the reviewer that a better positioning will help improve the paper. We will add the new experimental results above in the Appendix and the following discussion regarding the OOD issue and our methodology in Section 4.1 (in future versions of our manuscript as the deadline for uploading a revised manuscript during the response period has passed unfortunately).
>
> "The OOD issue -- when encountering state and actions unseen in the dataset -- is more nuanced in the offline safe RL setting due to the need to ensure safety in addition to policy performance. Our treatment of regarding state-action pairs that exceed the cost budget as OOD provides a means of incorporating safety constraints into the policy optimization process. This enables the learned policy to avoid unsafe regions of the state-action space even if they are within the dataset distribution, while also adapting to unseen and varying cost thresholds. As we will show in Section 4.2 and Section 4.3, this is done through a novel data augmentation and filtering process by a constraint-conditioned VAE and classifer, and an overestimation of the $Q_c$ values respectively. In Section 4.4, we offer further insights on how this overestimation of $Q_c$ can achieve a similar policy regularization effect as an underestimation of $Q_r$ (e.g., penalizing the value estimates for OOD actions in CQL) to mitigate the OOD issue in offline RL. A more explicit approach that distinguishes between the high-value and low-value state-actions when satisfying the cost constraints could be an interesting alternative to policy regularization for addressing the OOD issue, and we leave this exploration to future work."
>
> Regarding the case of "low-reward low-cost actions dominating the dataset", we would like to better understand the reviewer's perspective on how a high-value policy could be learned without introducing additional assumptions or interventions. Offline RL is inherently constrained by the dataset that it is trained on. If high-reward trajectories are absent or severely underrepresented, the algorithm lacks the information necessary to estimate the value of those trajectories accurately. Additionally, many offline RL algorithms, including ours, are designed to avoid OOD actions by remaining close to the behavior policies/data distribution, albeit through different mechanisms. A dataset dominated by low-reward, low-cost trajectories will bias the learned policy towards micmicing these trajectories rather than exploring higher-reward options (which may not even exist). Without any additional assumption or guidance, it is challenging to see how an offline method could extrapolate beyond the dataset to infer high-reward behaviors. In contrast, our previous experiment showed that we can extrapolate beyond "low-reward, low-cost" trajectories and learn a policy to perform in a regime of "high-reward, low-cost" even in the absence of such data in the offline dataset, because the "high-reward, high-cost" trajectories in the dataset can still provide useful learning signals (e.g., safe sub-trajectories with relatively high rewards).

---

> ### Comment · Reviewer_pD1A · 2024-12-02
>
> Thanks for the overarching paragraph, I am satisfied with this exposition.
>
> Regarding the low-value dominance, I meant these experiences account for a major proportion of data, not complete absence of good data. In this case naive behavior cloning or behavior regularization will not work very well as the authors suggested. There are some existing works focus on adjusting the degree to which the agent sticks to the behavior distribution. This can be done for example by in-sample softmax where the agent has an additional degree of freedom over actions to balance the similarity and OOD; or sparse update filtering to ensure only a small subset of actions with promising value get updated, see refs [1-3].
>
> References:\
> [1] the in-sample softmax for offline reinforcement learning\
> [2] offline rl with no ood actions: in-sample learning via implicit value regularization\
> [3] offline rl via Tsallis regularization

---

> > ### Author Response · Authors · 2024-12-02
> >
> > We would like to thank the reviewer for raising their score and for pointing out these references. Our experiments with IQL show that our proposed method is compatible with existing policy/value regularization techniques. We believe that adopting and integrating similar approaches, with the idea of adjusting the degree to which the agent adheres to behavior policies, as the reviewer mentioned, is an interesting direction — such as selectively updating policies toward certain low-risk, high-value actions within high-risk trajectories when dealing with datasets dominated by low-risk, low-value trajectories. Furthermore, exploring ways to handle varying dataset qualities in the offline safe RL setting could be an exciting avenue for future work. Once again, we appreciate the critical feedback and thoughtful discussions.

---

### Official Review · Reviewer_REk7 · 2024-11-04

**Soundness:** 3
**Presentation:** 3
**Contribution:** 3
**Rating:** 6
**Confidence:** 3

**Summary:**

In the context of offline safe reinforcement learning, this paper implements penalties for out-of-distribution (OOD) actions using a generative model, such as a conditional variational auto-encoder (CVAE).
Then, the authors propose a method to train a policy that conditionally adapts to different cost budgets using overestimated cost Q functions.

**Strengths:**

- The paper is well-written and easy to read.
- The proposed method shows the best performance.
- Through various ablation studies, the paper effectively shows that the proposed method is robust to OOD scenarios and can generate adaptive behaviors across different cost budgets.
- Section 5.3 provides experimental justification for the usage of CVAE and good visualization of the overestimated $Q_C$.

**Weaknesses:**

- In lines 46-53, the need for a varying threshold is highlighted, but this issue has been addressed in several studies [1,2].
It is crucial to motivate the proposed method by identifying shortcomings or potential improvements in the existing techniques on varying thresholds.

- In line 98 of Related Work, the statement “these methods overlook the dependence of actions on the constraints” seems to need to be toned down a bit. This seems obvious since the works mentioned by the authors are related to offline RL, not offline safe RL.

- The overview in Figure 1 needs to be further elaborated. It is difficult to understand what is being illustrated in the figure, as it does not clearly describe the inputs and outputs of each model and does not explain the learning flow. Also, the caption should include a description of each module and arrow.

- In lines 203-204, there is a need for a detailed explanation of the mechanism for generating out-of-distribution (OOD) data. According to the attached code, it appears that the $(s, a)$ pairs are generated through the CVAE using various Gaussian noise vectors $z$. Pairs, where the value of $h$ is less than $0.5$, are designated as OOD data. This process should be clearly explained in the paper.

- The TREBI paper [2] also deals with various cost budgets in offline environments. Unfortunately, the authors do not compare CCAC with this method, which deals with the same problem as the proposed method.

## References
- [1] Liu, Zuxin, et al. "Constrained decision transformer for offline safe reinforcement learning." International Conference on Machine Learning. PMLR, 2023.
- [2] Lin, Qian, et al. "Safe offline reinforcement learning with real-time budget constraints." International Conference on Machine Learning. PMLR, 2023.

**Questions:**

- It appears that the policy is trained through Equation 9. In traditional offline reinforcement learning methods, a penalty term is added to a policy loss to ensure that the actions of the policy do not deviate from those in the dataset. However, the proposed method addresses this by solely using the overestimated $Q_C$. Is this correct?

- In Table 1, COptiDICE shows significant constraint violations, but in its original paper, COptiDICE reported low violations in similar tasks.
An explanation for this discrepancy is needed.
In line 323, the authors mentioned that this is due to biased estimation. Still, this explanation is not entirely convincing since COptiDICE itself includes a component that trains the policy by overestimating the cost of OOD data.

---

> ### Author Response · Authors · 2024-11-16
> **Rebuttal by Authors**
>
> We appreciate the insightful feedback from the reviewer. Below, we detail our responses to each of the weaknesses and questions and the modified parts are colored in cyan in the updated manuscript.
>
> ### Weaknesses
> 1. In lines 46-53, the need for a varying threshold is highlighted, but this issue has been addressed in several studies [1,2]. It is crucial to motivate the proposed method by identifying shortcomings or potential improvements in the existing techniques on varying thresholds.
>
> We discuss the limitations of [1, 2] in the Related Work section in lines 82-87.
>
> 2. In line 98 of Related Work, the statement “these methods overlook the dependence of actions on the constraints” seems to need to be toned down a bit. This seems obvious since the works mentioned by the authors are related to offline RL, not offline safe RL.
>
> Thank you for pointing this out. We have rephrased the sentence to improve clarity in our revised manuscript.
>
> 3. The overview in Figure 1 needs to be further elaborated. It is difficult to understand what is being illustrated in the figure, as it does not clearly describe the inputs and outputs of each model and does not explain the learning flow. Also, the caption should include a description of each module and arrow.
>
> We have also added more details in Figure 1 in the revised manuscript.
>
> 4. In lines 203-204, there is a need for a detailed explanation of the mechanism for generating out-of-distribution (OOD) data. According to the attached code, it appears that the $(s, a)$ pairs are generated through the CVAE using various Gaussian noise vectors $z$. Pairs, where the value of $h$ is less than $0.5$, are designated as OOD data. This process should be clearly explained in the paper.
>
>
> Thank you for the suggestion. We have added more implementation details of our method in Appendix C.1 in the revised manuscript.
>
>
> 5. The TREBI paper also deals with various cost budgets in offline environments. Unfortunately, the authors do not compare CCAC with this method, which deals with the same problem as the proposed method.
>
> Thank you for highlighting this point. In addition to the existing discussion of TREBI in the Related Work section, we have now added a direct comparison with TREBI in Section 5 in the revised manuscript, using their official code and provided hyperparameters. The results are presented in Table 1 and Figure 3. TREBI has difficulty maintaining safety as it restricts its learned policy to remain close to the behavior policy (Eq. (3) and (6) in [2]).
>
> ### Questions
> 1. It appears that the policy is trained through Equation 9. In traditional offline reinforcement learning methods, a penalty term is added to a policy loss to ensure that the actions of the policy do not deviate from those in the dataset. However, the proposed method addresses this by solely using the overestimated $Q_c$. Is this correct?
>
> Yes, the policy is updated using Eq.(9) and no additional term is applied to restrict the actions of the policy to stay close to the datasets. Our method achieve this by only overestimating the $Q_c$ of OOD state-action paris (as explained in lines 254-258).
>
> 2. In Table 1, COptiDICE shows significant constraint violations, but in its original paper, COptiDICE reported low violations in similar tasks. An explanation for this discrepancy is needed. In line 323, the authors mentioned that this is due to biased estimation. Still, this explanation is not entirely convincing since COptiDICE itself includes a component that trains the policy by overestimating the cost of OOD data.
>
> We believe there are substantial differences between the original COptiDICE datasets and the DSRL datasets used in our work. In the COptiDICE paper [1], it’s noted that "the dataset is a mixture of low-reward-low-cost trajectories and high-reward-high-cost trajectories" (Section 4.2, last sentence of the second paragraph). By contrast, the DSRL dataset includes not only these trajectories but also low-reward-high-cost and high-reward-low-cost trajectories (as illustrated in Figures 10 and 11 of [2]). This broader diversity in the DSRL data adds complexity to estimating the datasets’ stationary distribution and the underlying behavior policies. Furthermore, Figure 3(c) in [1] demonstrates that COptiDICE struggles to maintain safety when the proportion of constraint-satisfying trajectories is low ($\beta = 0.2$). In the DSRL datasets, the proportion is even lower, averaging only 9.5% across the nine tasks we evaluated (as mentioned in the caption of Table 1). These factors contribute to the substantially higher safety violations of COptiDICE on the DSRL datasets. \
> [1] https://arxiv.org/pdf/2204.08957 \
> [2] https://arxiv.org/pdf/2306.09303

---

> ### Author Response · Authors · 2024-11-20
> **Reminder for Re-evaluation and Further Discussion**
>
> Dear reviewer,
>
> Thank you again for your thoughtful feedback. We hope our response has addressed your questions and concerns. If there’s anything further you’d like us to clarify, we’d be happy to do so. If our response has resolved your concerns, we kindly ask you to consider raising your score. Thank you!
>
> Best, \
> Authors

---

> > ### Comment · Reviewer_REk7 · 2024-11-25
> > **Reviewer Response**
> >
> > **1. Clarify the Motivation for Addressing Existing Limitations**
> >
> > The line 87, "Although they can generalize to different cost thresholds, they are sensitive to the rewards and costs they are conditioned on," does not clearly convey the specific shortcomings of existing methods. (what does "sensitive" mean?)
> > Furthermore, intuitive motivation is needed to determine how and why the proposed approach improves upon these methods.
> > Providing a more detailed explanation of the limitations of the existing methods and explicitly highlighting the key differences of your approach would make the motivation more compelling.
> >
> > **2. Enhance Figure 1 Description**
> >
> > It would be helpful to include the explanation, "Among the data generated by CVAE, those with a classifier score above 0.5 are added as OOD data," in the description of Figure 1 since the role of the classifier is not clear from the current caption.
> >
> > If the author addresses these two points, I'd be happy to raise the score.

---

> ### Author Response · Authors · 2024-11-27
>
> We thank the reviewer for the addtional feedback.
>
> 1. Clarify the Motivation for Addressing Existing Limitations.
>
> > "Although they can generalize to different cost thresholds, they are sensitive to the rewards and costs they are conditioned on,"
>
> Here, "sensitive" means that the performance of those transformer-based methods is sensitive to the input cost and reward tokens especially those that are not present in the dataset, e.g. performance degrades for combinations of high rewards and low costs that do not exist in the dataset. The reason is that these methods learn policies via supervised learning, i.e. micmicing what we have in the dataset.
>
> > "a more detailed explanation of the limitations of the existing methods"
>
> In many real-world scenarios, the dataset can be collected from multiple behavior policies that have diverse behaviors based on different cost thresholds. Our goal is to learn policies that can automatically adapt to these thresholds. The key differece between our method and existing methods is that we learn constraint-conditioned critics, $Q_r(a| s, \kappa)$ and $Q_c(a | s, \kappa)$ and policy $\pi(a | s, \kappa)$, while existing value-based methods typically learns policy based only on states, $\pi(a | s)$ (Section 4.1). The advantage of constraint-conditioning is that it can better model the behavior policies of the dataset and offer the flexibility to generalize to varying constraint thresholds. In contrast to supervised-learning-based methods, we learn the policy through policy evaluation and improvement (Eq.(6-9)) rather than mimicing the behaviors in the dataset as the dataset contains (probably with a majority of) sub-optimal trajectories. Moreover, existing methods struggle to generalize to new or unseen cost thresholds. We have updated our manuscript correspondingly in the Introduction and Related Work sections.
>
> 2. Enhance Figure 1 Description.
>
> The caption of Figure 1 has been updated to include the role of the classifier.

---

> > ### Comment · Reviewer_REk7 · 2024-11-28
> > **Reviewer Response**
> >
> > Thanks for the clarification.
> > The modified motivation parts are persuasive now.
> > I have raised my score.

---

### Official Review · Reviewer_3i9u · 2024-11-04

**Soundness:** 3
**Presentation:** 3
**Contribution:** 2
**Rating:** 6
**Confidence:** 3

**Summary:**

The paper presents a constraint-conditioned actor-critic method, which combines a constraint-conditioned variational autoencoder (CVAE) and a constraint-conditioned classifier to learn this relationship.

**Strengths:**

1. The paper is well-written and easy to follow.

2. The visualization of the offline datasets is quite intuitive, and the experimental design, along with the ablation study, adds significant value to the research.

3. The method can learn safe and high-reward policies from offline datasets, demonstrating better adaptability and robustness to distribution shifts.

**Weaknesses:**

1. The experimental environments are relatively simple and do not include more complex tasks, such as humanoid. In the experimental section, the authors use only three seeds, which I believe is insufficient. If possible, could you provide additional results using more seeds (at least five)?

2. Although the algorithm demonstrates performance compared to the baselines while maintaining safety, Figure 9 suggests that the algorithm's sample efficiency is not good.

**Questions:**

1. The training process of CCAC appears to separate the training of the CVAE and the policy (optional once). Why not consider continuing the training of the CVAE during the policy training process?

2. CCAC demonstrates zero-shot adaptation to different cost budgets. In addition to empirical evidence, could you provide some intuitive explanations for this capability?

---

> ### Author Response · Authors · 2024-11-16
> **Rebuttal by Authors**
>
> ## Part I
> We thank the reviewer for your thoughtful comments and constructive feedback. We address each weakness and question as follows.
>
> ### Weaknesses
> 1. The experimental environments are relatively simple and do not include more complex tasks, such as humanoid. In the experimental section, the authors use only three seeds, which I believe is insufficient. If possible, could you provide additional results using more seeds (at least five)?
>
> We would like to clarify our experimental choices. Regarding the exvironments, the DSRL benchmark does not provide a dataset for Humanoid tasks. In our experiments, we included Ant-Vel and Ant-Circle, which are the most complex locomotion tasks available within the benchmark. These tasks have high state (Ant-Vel: 27, Ant-Circle: 34) and action dimensions (8 for both) [1]. Regarding the number of seeds, we followed the evaluation protocol in DSRL and related works that benchmark on DSRL. Studies like [2], [3], and [4] also use 3 seeds for evaluation. To further alleviate the reviewer's concern, we have also trained and evaluated our method using two more seeds and the results (5 seeds) are shown in the table below, where $\Delta = |$CCAC(5 seeds) - CCAC(3 seeds)$|$. We can observe that there is no significant difference in performance.
>
> |         Task         |       Metric      |      CCAC(3 seeds)      |     CCAC(5 seeds)     | $\Delta$ |
> |:--------------------:|:-----------------:|:-----------------------:|:---------------------:|:--------:|
> |       Ball-Run       | Reward $\uparrow$ | 0.97 $\pm$ 0.01 | 0.96 $\pm$ 0.01 |   0.01   |
> |                      | Cost $\downarrow$ | 0.27 $\pm$ 0.19 | 0.36 $\pm$ 0.27 |   0.09   |
> |        Car-Run       | Reward $\uparrow$ | 0.95 $\pm$ 0.04 | 0.93 $\pm$ 0.04 |   0.02   |
> |                      | Cost $\downarrow$ | 0.19 $\pm$ 0.27 | 0.13 $\pm$ 0.23 |   0.06   |
> |      Ant-Circle      | Reward $\uparrow$ | 1.01 $\pm$ 0.26 | 1.04 $\pm$ 0.21 |   0.03   |
> |                      | Cost $\downarrow$ | 0.55 $\pm$ 1.57 |  0.4 $\pm$ 1.32 |   0.15   |
> |      Ball-Circle     | Reward $\uparrow$ | 0.87 $\pm$ 0.03 | 0.88 $\pm$ 0.03 |   0.01   |
> |                      | Cost $\downarrow$ |  0.0 $\pm$  0.0 |  0.0 $\pm$ 0.0  |    0.0   |
> |      Car-Circle      | Reward $\uparrow$ | 0.85 $\pm$ 0.04 | 0.85 $\pm$ 0.04 |    0.0   |
> |                      | Cost $\downarrow$ | 0.73 $\pm$ 1.95 | 0.72 $\pm$ 2.04 |   0.01   |
> |     Drone-Circle     | Reward $\uparrow$ | 0.82 $\pm$ 0.11 | 0.83 $\pm$ 0.09 |   0.01   |
> |                      | Cost $\downarrow$ | 0.07 $\pm$ 0.54 | 0.06 $\pm$ 0.45 |   0.01   |
> |     Ant-Velocity     | Reward $\uparrow$ |  0.9 $\pm$ 0.05 | 0.91 $\pm$ 0.07 |   0.01   |
> |                      | Cost $\downarrow$ | 0.58 $\pm$ 0.15 |  0.57 $\pm$ 0.1 |   0.01   |
> | HalfCheetah-Velocity | Reward $\uparrow$ | 0.96 $\pm$ 0.04 | 0.98 $\pm$ 0.03 |   0.02   |
> |                      | Cost $\downarrow$ |  0.79 $\pm$ 0.2 | 0.83 $\pm$ 0.12 |   0.04   |
> |    Hopper-Velocity   | Reward $\uparrow$ | 0.89 $\pm$ 0.02 | 0.88 $\pm$ 0.08 |   0.01   |
> |                      | Cost $\downarrow$ | 0.32 $\pm$ 0.23 | 0.42 $\pm$ 0.23 |    0.1   |
> |        Average       | Reward $\uparrow$ | 0.91 $\pm$ 0.11 | 0.92 $\pm$ 0.11 |   0.01   |
> |                      | Cost $\downarrow$ | 0.39 $\pm$ 0.94 |  0.4 $\pm$ 0.92 |   0.01   |
>
> [1] https://arxiv.org/pdf/2306.09303 \
> [2] https://arxiv.org/abs/2401.10700 \
> [3] https://arxiv.org/pdf/2407.14653 \
> [4] https://proceedings.mlr.press/v238/hong24a/hong24a.pdf
>
> 2. Although the algorithm demonstrates performance compared to the baselines while maintaining safety, Figure 9 suggests that the algorithm's sample efficiency is not good.
>
> We acknowledge that CCAC takes more steps to converge in certain environments, such as Car-Run and Drone-Circle. However, it is important to note that we are working in the offline safe RL setting where training occurs solely on a fixed offline dataset. Taking more steps to converge also does not imply additional safety violations as there is no interaction with the environment during training. Furthermore, while sample efficiency is valuable, a method that achieves efficiency at the expense of safety may be less meaningful. Figure 9 shows that although some baselines converge faster, such as FISOR in Car-Run and BCQ-Lag in Drone-Circle, they either fail to achieve high rewards or cannot maintain safety after convergence. In contrast, our method consistently achieves both high reward and safety after convergence. Finally, the number of iterations is not necessarily a good reflection of runtime or computational cost (if this is what the reviewer is concerned about), methods like FISOR, CDT and TREBI are more computationally expensive in each iteration due to the use of Transformer-based models or diffusion models.

---

> ### Author Response · Authors · 2024-11-16
> **Rebuttal by Authors**
>
> ## Part II
>
> ### Questions
> 1. The training process of CCAC appears to separate the training of the CVAE and the policy (optional once). Why not consider continuing the training of the CVAE during the policy training process?
>
> The training process of CVAE and the policy are indeed seperate as we mentioned in lines 264-269. We would like to ehphasize that policy training does not impact CVAE training (as Eq.(3-4) shows, CVAE training is independent of the policy), while training CVAE does affect the policy training (as Eq.(5-9) show, policy training involves CVAE to generate OOD state-action pairs). Thus, we choose to train the CVAE until convergence before starting policy training since the quality of CVAE and classifier is critical to accurately distinguish OOD state-action paris which then help guide the policy learning. Although one could continue CVAE training during policy training (as noted in line 14 of Algorithm 1 in Appendix C.1), since the CVAE has already converged, further training has minimal impact on policy performance. Thus, we opted not to continue CVAE training during policy training.
>
>
> 2. CCAC demonstrates zero-shot adaptation to different cost budgets. In addition to empirical evidence, could you provide some intuitive explanations for this capability?
>
> The key to our method’s zero-shot adaptation to different cost budgets lies in its use of a constraint-conditioned policy. Unlike most offline safe RL algorithms, which learn a policy $\pi(a | s)$ dependent only on the state, our approach learns a constraint-conditioned policy $\pi(a | s, \kappa)$. This design allows our policy to adapt actions based on the given cost budget, even for the same state. In contrast, a standard $\pi(a | s)$ policy lacks this flexibility, as it does not account for the current cost budget. This conditioning enables our policy to generalize effectively across varying cost budgets.

---

> > ### Comment · Reviewer_3i9u · 2024-11-25
> >
> > Thank you for your detailed response to my review. The response has addressed most of my questions.
> > I agree with reviewer pD1A's concerns regarding the issues with OOD and OOB data. But I am inclined to keep my original score.

---

> ### Author Response · Authors · 2024-11-21
> **Reminder for Re-evaluation and Further Discussion**
>
> Dear reviewer,
>
> Thanks again for your valuable comments. We hope our new response has addressed your concerns, and we would greatly appreciate it if you could re-evaluate our submission based on the response, or let us know whether you have any other questions. Thanks!
>
> Best, \
> Authors

---

> ### Author Response · Authors · 2024-11-26
>
> Thank you for reviewing our response and for your feedback. We are glad to hear that our response has addressed most of your questions. Regarding reviewer pD1A's concerns about OOD and OOB data, we have included further clarifications and additional experiments (please refer to response "Additional clarifications and results regarding the OOD issue" to reviewer pD1A). These results demonstrate that the concerns raised stemmed from a misunderstanding of how our method works and are not supported by the analysis and evidence presented in the paper and in our response.

---

### Author Response · Authors · 2024-12-02
**General Response**

We sincerely thank all the reviewers for their valuable feedback and thoughtful discussions, which have greatly contributed to improving the paper and identifying promising directions for future work. We are glad that the reviewers found the paper well-structured and appreciated the comprehensive evaluation results which show that the proposed CCAC method outperforms existing methods in learning safe, high-reward policies and adapting to varying constraint thresholds. We hope we have adequately addressed all the concerns and will update the paper accordingly. We would be happy to provide further clarifications if needed.

---

### Meta-Review · Area_Chair_qw2f · 2024-12-23

**Metareview:**

In this paper, the authors propose an offline safe RL algorithm, called constraint conditioned actor-critic (CCAC). CCAC is designed to learn safe policies under different constraint budgets, while controlling the distribution shift issue that exist in the offline RL setting. CCAC uses a generative model, such as a conditional variational auto-encoder (CVAE), and a classifier to detect out-of-distribution state-action pairs, to implement penalties for OOD actions. Using this framework, it executes actor-critic updates and maintains a constraint-aware critic and policy. The authors provide theoretical justification and comprehensive empirical investigation (extensive ablation studies) on the DSRL benchmark.

(+) The paper is well-written and easy to follow.
(+) The authors properly show the robustness of their approach to OODs and its ability to adapt to different constraint budgets using comprehensive ablation studies.

Despite several weaknesses listed by the reviewers, such as
(-) The sample inefficiency of the proposed algorithm.
(-) Some missing baselines, such as TREBI.
(-) Some strong claims and statements that need to be toned down.
(-) And most importantly, the lack of clarity between out-of-distribution (OOD) and out-of-budget (OOB) state-action pairs. I agree with Reviewer pD1A that it does not make sense to to assume all OOD state-action pairs are unsafe.

the paper has a comprehensive experiments and overall is well-executed, thus the community, especially practitioners, could find it useful. I strongly recommend that the authors take the reviewers' comments, especially those in the discussions (e.g., with Reviewer pD1A), into account and revise their paper accordingly before it gets published.

**Additional Comments On Reviewer Discussion:**

The authors managed to clarify certain issues pointed out by the reviewers, and thus, some of them raised their scores. I hope the authors pay special attention to their discussions with the reviewers (in addition to the reviews) in preparing the final draft of their paper (emphasized in the meta-review).

---

### Decision · Program_Chairs · 2025-01-22

Accept (Poster)